# Crystal structure of intraflagellar transport protein 80 reveals a homo-dimer required for ciliogenesis

**Michael Taschner[1†‡], Anna Lorentzen[1†], André Mourão[2†], Toby Collins[3†], Grace M Freke[3], Dale Moulding[4], Jerome Basquin[5], Dagan Jenkins[3*], Esben Lorentzen[1*]**

[1]Department of Molecular Biology and Genetics, Aarhus University, Aarhus, Denmark; [2]Institute of Structural Biology, Helmholtz Zentrum München, Neuherberg, Germany; [3]Genetics and Genomic Medicine, University College London, London, United Kingdom; [4]Developmental Biology and Cancer Programmes, Great Ormond Street Institute of Child Health, University College London, London, United Kingdom; [5]Department of Structural Cell Biology, Max-Planck-Institute of Biochemistry, Martinsried, Germany

**\*For correspondence:**
d.jenkins@ucl.ac.uk (DJ);
el@mbg.au.dk (EL)

[†]These authors contributed equally to this work

**Present address:** [‡]Department of Fundamental Microbiology, University of Lausanne, Lausanne, Switzerland

**Competing interests:** The authors declare that no competing interests exist.

**Abstract** Oligomeric assemblies of intraflagellar transport (IFT) particles build cilia through sequential recruitment and transport of ciliary cargo proteins within cilia. Here we present the 1.8 Å resolution crystal structure of the *Chlamydomonas* IFT-B protein IFT80, which reveals the architecture of two N-terminal β-propellers followed by an α-helical extension. The N-terminal β-propeller tethers IFT80 to the IFT-B complex via IFT38 whereas the second β-propeller and the C-terminal α-helical extension result in IFT80 homo-dimerization. Using CRISPR/Cas to create biallelic *Ift80* frameshift mutations in IMCD3 mouse cells, we demonstrate that IFT80 is absolutely required for ciliogenesis. Structural mapping and rescue experiments reveal that human disease-causing missense mutations do not cluster within IFT80 and form functional IFT particles. Unlike missense mutant forms of IFT80, deletion of the C-terminal dimerization domain prevented rescue of ciliogenesis. Taken together our results may provide a first insight into higher order IFT complex formation likely required for IFT train formation.
DOI: https://doi.org/10.7554/eLife.33067.001

## Introduction

Cilia and flagella are eukaryotic cellular organelles important for motility, sensory reception, and developmental signalling (*Ishikawa and Marshall, 2011*), and defects in ciliary structure and/or function result in human pathologies collectively known as 'ciliopathies' (*Fliegauf et al., 2007*). Cilia are found on most cells in mammals, as well as on unicellular organisms such as *Chlamydomonas reinhardtii* (*Cr*) and *Trypanosoma brucei (Tb)* that serve as model organisms for ciliary studies (*Vincensini et al., 2011*). With only very few exceptions, cilia are built by intraflagellar transport (IFT), the bi-directional movement of proteinaceous material sandwiched between the microtubule (MT)-based axoneme and the ciliary membrane (*Kozminski et al., 1993*; *Rosenbaum and Witman, 2002*). The process of IFT in *Chlamydomonas* requires the 22 subunit IFT particle that associates with the hetero-trimeric kinesin 2 or cytoplasmic dynein 2 motors for anterograde (ciliary base to tip) or retrograde (ciliary tip to base) transport of cargo proteins, respectively. In *C. elegans* sensory cilia the situation is more complex, as a homo-dimeric kinesin 2 (OSM-3) cooperates with heterotrimeric kinesin 2 to drive anterograde IFT (*Snow et al., 2004*; *Prevo et al., 2015*).

The IFT particle consists of a six subunit IFT-A and a 16 subunit IFT-B complex, with the latter further divided into a 10 subunit IFT-B1 and a 6-subunit IFT-B2 sub-complex (*Taschner et al., 2016*). Whereas IFT-A and IFT-B complexes dissociate even at low salt concentration when isolated from *Cr* flagella (*Cole et al., 1998*), IFT-B1 and IFT-B2 sub-complexes associate strongly to form a stable IFT-B complex that can be reconstituted from recombinantly produced subunits (*Taschner et al., 2016*). Electron tomographic reconstructions of IFT material in situ reveal that IFT particles organize into long strings known as IFT trains in *Chlamydomonas* flagella (*Pigino et al., 2009*; *Vannuccini et al., 2016*). IFT trains have also been observed in *Trypanosoma* (*Absalon et al., 2008*) and in some cases in human primary cilia (*Rogowski et al., 2013*). IFT proteins are sequentially recruited at the basal body to form IFT trains that are loaded with tubulin cargo shortly before departure (*Wingfield et al., 2017*). An elegant study coupling total internal reflection fluorescence (TIRF) and electron microscopy recently revealed that anterograde and retrograde IFT trains travel on different tubules of the axonemal MT-doublets in *Chlamydomonas* flagella, providing an explanation for the absence of head-to-head collisions of IFT trains (*Stepanek and Pigino, 2016*). Two morphologically different types of short (~200 nm in length, 16 nm repeat) arrays with IFT particle volumes of ~5000 nm$^3$ correspond to the moving anterograde and retrograde trains (*Pigino et al., 2009*; *Vannuccini et al., 2016*). Additionally, a longer (~650 nm in length, 40 nm repeat) IFT train type with two IFT particles related by 2-fold symmetry and a volume of ~10000 nm$^3$ was identified as a 'standing' train not moving in either direction (*Pigino et al., 2009*; *Stepanek and Pigino, 2016*). It is currently not known how IFT particles associate to form either of these different types of IFT trains.

Since the discovery of the IFT complex in *Chlamydomonas* two decades ago, significant progress has been made in understanding its composition from studies of material obtained directly from *Cr* flagella (*Cole et al., 1998*; *Piperno and Mead, 1997*; *Lucker et al., 2005*; *Behal and Cole, 2013*; *Richey and Qin, 2012*), as well as from work using recombinantly expressed proteins (*Taschner et al., 2016*; *Lucker et al., 2010*; *Taschner et al., 2011*; *Taschner et al., 2014*). Furthermore, an increasing number of crystal structures provide us with a detailed view of several parts of this sophisticated transport machinery (*Taschner et al., 2016*; *Taschner et al., 2014*; *Bhogaraju et al., 2011*; *Bhogaraju et al., 2013*). However, none of these studies have so far provided any clues about how IFT complexes form larger assemblies, which presumably relies on the ability of one or several IFT proteins to oligomerize.

The hexameric IFT-B2 complex was recently identified in several studies (*Taschner et al., 2016*; *Boldt et al., 2016*; *Katoh et al., 2016*). Missense mutations in *IFT80*, which encodes an IFT-B2 subunit, have been identified in individuals with the skeletal ciliopathies Jeune asphyxiating thoracic dystrophy (JATD) as well as Verma-Naumoff syndrome (*Baujat et al., 2013*; *Beales et al., 2007*; *Cavalcanti et al., 2011*), but it is still unclear how these alterations affect protein function. Previous gene knockdown studies of *Ift80*, which have used RNA interference in mammalian cell culture or conditional gene-targeting in mice, have reported a quantitative reduction, but incomplete loss, of axonemal acetylated tubulin of primary and motile cilia (*Beales et al., 2007*; *Yuan and Yang, 2015*; *Yuan et al., 2016*; *Yang and Wang, 2012*; *Wang et al., 2013*). A constitutional hypomorphic *Ift80* gene-trap mouse model faithfully recapitulated features of JATD without affecting cilia formation (*Rix et al., 2011*). It is therefore unclear whether IFT80 is absolutely required for IFT mediated recruitment and transport of ciliary cargo. It has also been speculated that *IFT80* missense mutations may be hypomorphic although this remains to be proven.

Here, we present the crystal structure of CrIFT80 to show an unusual N-terminal double β-propeller followed by α-solenoid structure. Despite a shared domain organization with vesicle coatomer subunits (*Jékely and Arendt, 2006*; *van Dam et al., 2013*), the 3D organization of IFT80 domains is remarkably unique and allows for homo-dimerization of the protein. We show that the N-terminal β-propeller tethers IFT80 to the IFT complex via direct binding to the calponin-homology (CH) domain of IFT38 (also known as CLUAP1, FAP22, DYF-3, Qilin or PIFTA1) whereas the C-terminal domain is required for IFT80 dimerization. Using gene-editing to knockout *Ift80* in IMCD3 mouse cells, we demonstrate that IFT80 is absolutely required for initiation of the ciliary axoneme. Using rescue experiments, we show that missense mutations in *IFT80* retain significant biochemical function, and that truncations where dimerization is abolished are unable to rescue cilium formation. This suggests that higher order IFT complex formation via IFT80 is functionally important for ciliogenesis, and that disease-causing missense mutations operate via a distinct mechanism.

# Results

## Purification of IFT80 constructs

IFT80 is predicted to contain two N-terminal WD-40 β-propellers followed by α-solenoid structure (*Taschner et al., 2012*). This domain architecture is similar to vesicle coatomer proteins such as the coatomer subunits α and β' of the COPI coat (*Lee and Goldberg, 2010*), suggesting that IFT80 evolved from a proto-coatomer subunit in an ancestral cell (*Jékely and Arendt, 2006*; *van Dam et al., 2013*). Given the increased endo- and exocytosis taking place at membrane invaginations known as the ciliary pocket (*Benmerah, 2013*; *Pedersen et al., 2016*), located at the base of the cilium where IFT trains enter and exit cilia, it is not inconceivable that IFT proteins such as IFT80 may be directly involved in these processes. Indeed, IFT proteins have been visualized on vesicles destined for the cilium in *Chlamydomonas* (*Wood and Rosenbaum, 2014*) and membrane shedding from cilia has also been observed for a number of organisms and cell types (*Wood and Rosenbaum, 2015*; *Nager et al., 2017*; *Wang et al., 2014*). However, due to the absence of a high-resolution IFT80 structure it has so far been unclear if this similarity in predicted domain composition translates into similar 3D-architectures and functions of these proteins. We set out to shed light on this issue by screening several CrIFT80 constructs for crystallization. In addition to full-length (fl, residues 1–

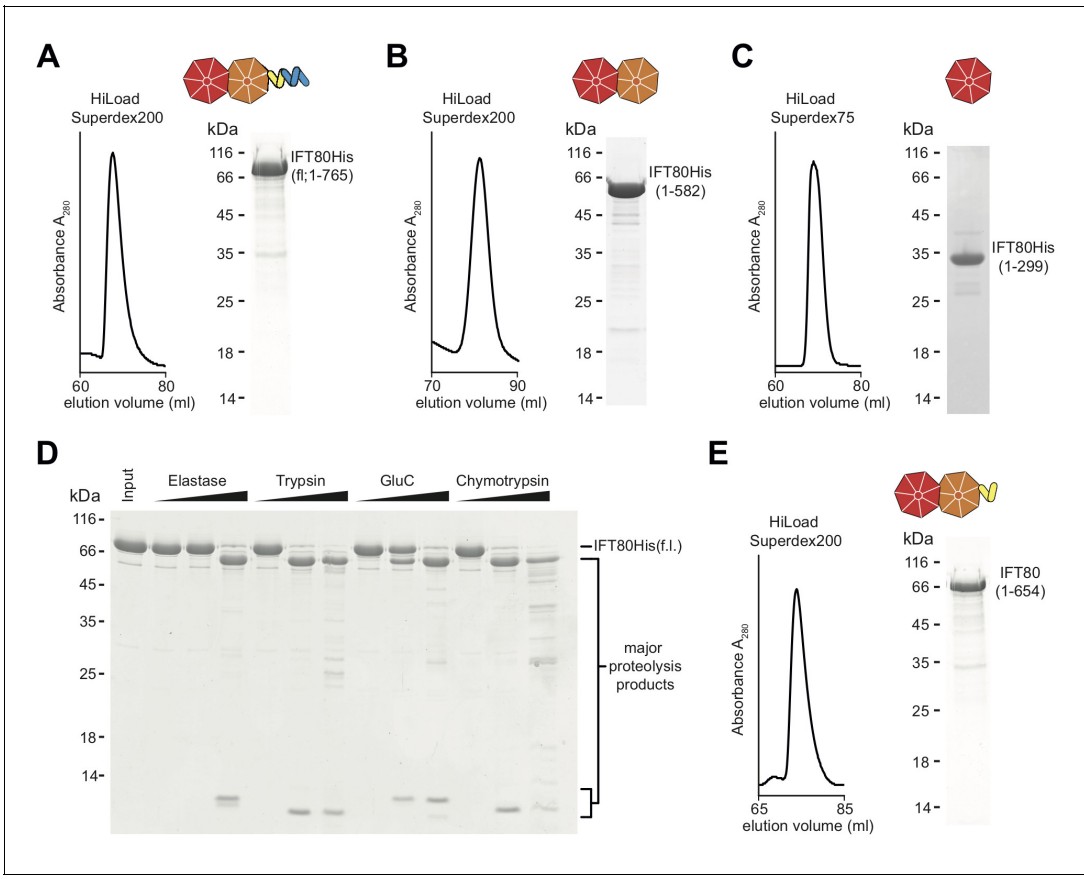

**Figure 1.** Purification of CrIFT80 constructs. (A–C) SEC profiles (left) and corresponding Coomassie-stained SDS-PAGE pictures of CrIFT80 fragments designed based on domain prediction. A schematic representation of the proteins is shown above the gel picture for full-length CrIFT80 (residues 1–765) (A), CrIFT80 BP1-BP2 (residues 1–582) (B), and CrIFT80 BP1 (residues 1–299) (C). (D) Coomassie-stained SDS-PAGE gel showing the limited proteolysis results for full-length CrIFT80 treated with the proteases Elastase, Trypsin, GluC, and Chymotrypsin. The positions of the full-length protein as well as for the major proteolysis products are indicated. All proteases lead to the cleavage of the protein into stable fragments of similar sizes. (E) Large-scale proteolysis of CrIFT80 with the protease GluC followed by SEC (profile on the left) led to the identification by mass-spectrometry of a fragment encompassing residues 1–654. The Coomassie-stained SDS-PAGE gel on the right shows the purity of this product, and the schematic representation on top shows that it is composed of BP1 and BP2 followed by a short stretch of alpha-solenoid structure.
DOI: https://doi.org/10.7554/eLife.33067.002

765) IFT80 of *Cr* (*Figure 1A*), we purified constructs containing both β-propellers (BP1-BP2, residues 1–582), as well as only the most N-terminal β-propeller (BP1, residues 1–299) (*Figure 1B–C*). None of the constructs could be expressed in *E.coli* in a soluble form and were thus produced in insect cells. The second β-propeller (BP2, residues 300–586) was insoluble even when recombinantly expressed in insect cells and could thus not be purified. In addition to these constructs designed based on domain predictions, we used limited proteolysis of fl IFT80 to determine additional stable fragments experimentally. This analysis resulted in the production of several fragments with molecular weights of ~65 kDa (*Figure 1D*), and further purification by size exclusion chromatography (SEC) of the IFT80 fragment generated by the protease GluC (*Figure 1E*) led to the identification of residues 1–654 by mass spectrometry (data not shown), consistent with proteolytic cleavage after residue E654. This fragment of IFT80 contains the two β-propellers followed by four of the eleven predicted C-terminal α-helices. Interestingly, expression of this truncated construct in insect cells did not yield soluble protein (data not shown), and it could only be obtained by proteolysis of full-length IFT80. The constructs shown in *Figure 1* facilitated a comprehensive analysis of the structure and function of IFT80.

## The IFT80 crystal structure reveals a unique domain arrangement different from vesicle coating subunits

All CrIFT80 protein constructs shown in *Figure 1* were screened for crystallization, resulting in an initial crystal hit and experimental electron density at 3 Å resolution for the proteolysed IFT80$_{1-654}$ fragment, which allowed for the construction of a partial model (*Table 1* and data not shown). When crystals were later obtained for fl IFT80 diffracting to 1.8 Å resolution, this partial model was used in molecular replacement to finalize the IFT80 structure (*Figure 2A* and *Table 1*). In the crystal structure, IFT80 adopts the fold of two consecutive seven bladed β-propellers followed by an alpha-solenoid extension of 4 α-helices. These helices organize into two tetratricopeptide (TPR)-like repeats, and are held in position via interactions with an extension of blade 6 of BP2. This extension contains an insertion of two α-helices, which are themselves held in place by largely hydrophobic contacts with BP2 blade 5 (*Figure 2A*). Although the fl IFT80 protein was crystallized (verified by mass spectrometry), the C-terminal 126 residues have no visible electron density and are presumably disordered in the crystal. Our finding that this region is efficiently removed by incubation with all four proteases tested (*Figure 1D*) supports this assumption. The missing residues are predicted to adopt the structure of seven α-helices and are connected to the four visible helices by a well-conserved 15 amino acid long linker region (*Figure 2—figure supplement 1*). In the IFT80 structure, BP1 and BP2 interact intimately via numerous loop regions connecting the blades. BP1 adopts the fold of a canonical seven bladed β-propeller where the N-terminal residues contribute the first β-strand to blade 7, effectively closing the β-propeller by a zipper-like mechanism (*Figure 2B*). BP1 of IFT80 is thus similar to a number of previously determined structures of β-propellers, and displays particularly high structural similarity to the N-terminal β-propellers of β'-COP, β-COP and α-COP, which are coatomer subunits important in the formation of COP-coated vesicles (*Lee and Goldberg, 2010*). Although BP1 of IFT80 and β'-COP only share ~15% sequence identity, the domains superimpose well with a root-mean-square-deviation (rmsd) of 2.2 Å over ~300 residues (*Figure 2D*). This high degree of structural similarity is consistent with the notion that IFT80 evolved from a coatomer-like progenitor (*Jékely and Arendt, 2006*; *van Dam et al., 2013*). The high degree of structural similarity between IFT80 and coatomer subunits is, however, limited to BP1. BP2 of IFT80 is non-canonical and differs in two important aspects from BP2 of β'-COP. Firstly, in the IFT80 structure, the most C-terminal β-strand of BP1 connects to BP2 as one long continuous β-strand (*Figure 2A*). This is quite different from β'-COP where BP1 and BP2 are connected via a loop (*Figure 2D*). This structural divergence is important as it positions BP2 differently relative to BP1 in the two structures (*Figure 2D*). Secondly, the extended loop that connects β-strands 3 and 4 of blade 1, and interacts with the last strand in blade seven to effectively close BP2 in β'-COP, is missing in IFT80 (compare *Figure 2C–2E*). This structural difference results in an unusual open conformation of BP2 of IFT80 where blades 1 and 7 do not interact (*Figure 2C*). Because of the very different positions of BP2 in IFT80 and β'-COP, the C-terminal α-helical extensions are found on opposite sides in the structures (*Figure 2D*). Thus, despite the similar domain composition of IFT80 and coatomer subunits, the relative 3D orientation of domains is remarkably different. Given the structural divergence, IFT80 is unlikely to function directly in the coating of vesicles destined for the cilium.

**Table 1.** X-ray data collection and refinement statistics

| | CrIFT80$_{1-654}$ EMP soak | CrIFT80 (fl) native |
|---|---|---|
| PDB code | | 5N4A |
| Data collection and scaling | Anomalous signal from Hg to ~ 4 Å | |
| Wavelength (Å) | 1.0077 | 1.0000 |
| Resolution range (Å) | 50–2.98 (3.13–2.98) | 47–1.79 (1.90–1.79) |
| Space group | P3112 | C2221 |
| Unit cell (Å) | a = 139.2 b = 139.2 c = 147.7 α = β = 90 γ = 120.0 | a = 82.8 b = 195.4 c = 117.5 α = β = γ = 90.0 |
| Total reflections | 1356277 (160974) | 718465 (109362) |
| Unique reflections | 33336 (4251) | 173548 (27521) |
| Multiplicity | 40.7 (37.9) | 6.6 (6.3) |
| Completeness (%) | 99.5 (96.5) | 99.5 (97.3) |
| Mean I/sigma(I) | 19.8 (2.8) | 11.3 (1.0) |
| R-merge | 0.225 (1.82) | 0.095 (1.40) |
| CC½ | 0.999 (0.817) | 0.998 (0.435) |
| Refinement | | |
| Number of reflections | | 173479 (27521) |
| Number of atoms (non-solvent) | | 5286 |
| Water (solvent) | | 768 |
| R-work | | 0.175 (0.324) |
| R-free | | 0.208 (0.351) |
| Ramachandran favoured (%) | | 97.14 |
| Ramachandran allowed (%) | | 2.81 |
| Ramachandran outliers (%) | | 0.15 |
| RMS bonds (Å) | | 0.0074 |
| RMS angles (°) | | 1.1 |
| Rotamer outliers (%) Clashscore | | 1.2 13.2 |
| Average B-Factors (Å$^2$) | | 28.2 |

DOI: https://doi.org/10.7554/eLife.33067.005

## BP1 tethers IFT80 to the IFT complex through a high affinity interaction with IFT38

How is IFT80 incorporated into the IFT-B2 complex? We previously showed that IFT80 interacts with the calponin-homology (CH) domain (residues 1–126) of IFT38 (*Taschner et al., 2016*), but the domain of IFT80 responsible is not known. Plotting the degree of amino acid conservation onto the IFT80 structure presented here revealed a number of conserved surface patches that may serve as interfaces for protein-protein interaction (*Figure 3A*). Most prominent is the conservation of the N-terminal surface of BP1, which harbours several invariant residues. To elucidate which domain of IFT80 is responsible for the interaction with IFT38 we used GST-tagged IFT38 CH-domain to pull-down fl, BP1-BP2 or BP1 of IFT80 (*Figure 3B*). The result of this experiment shows that IFT80 BP1 is sufficient for complex formation with the IFT38 CH-domain. Indeed, a complex of IFT80(BP1) and the IFT38 CH-domain could be reconstituted and purified by SEC (*Figure 3C*). Isothermal titration calorimetry (ITC) of the CH-domain of IFT38 and the BP1-BP2 construct of IFT80 (this construct was chosen as it displayed the best behaviour in ITC assays) revealed that IFT80-IFT38 is a stoichiometric

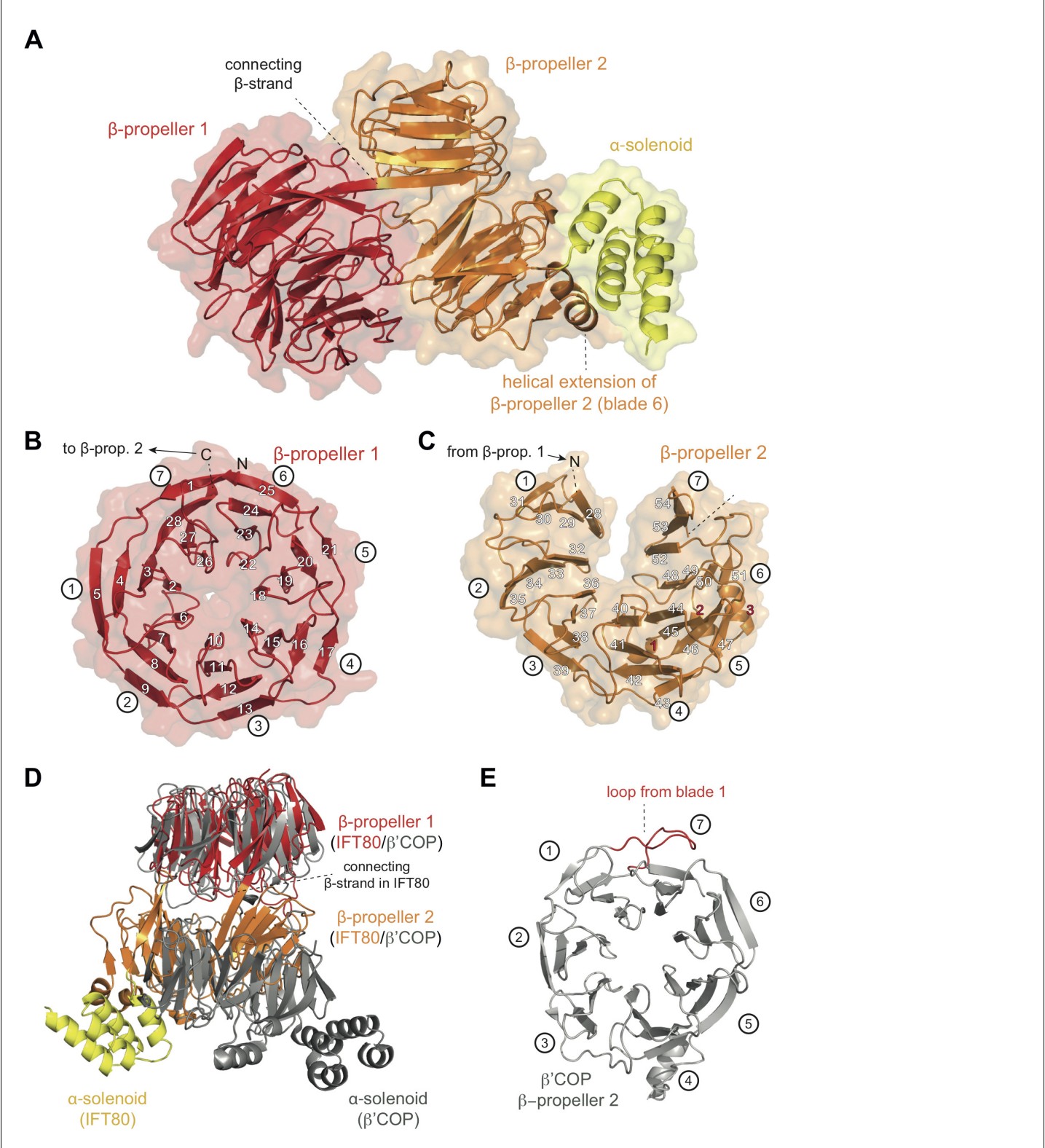

**Figure 2.** Crystal structure of the IFT80 protein. (A) Structural overview showing the overall domain organization of CrIFT80. It contains two N-terminal β-propellers (coloured in red and orange) connected by a long strand and interacting with each other via several loops. A helical extension in β-propeller two keeps the C-terminal α-solenoid region (coloured in yellow) in position. (B) β-propeller one displays a canonical fold of a 7-bladed WD40 β-propeller. The most N-terminal β-strand forms the outermost strand of blade 7, effectively closing the ring-shaped structure by linking blade 7 to blade 1. Individual β-strands are numbered and the blade numbers are indicated in white circles. (C) β-propeller two displays an unusual open

*Figure 2 continued on next page*

*Figure 2 continued*

conformation where the incomplete blade seven lacks the outermost β-strand. Individual β-strands are numbered and the blade numbers are indicated in white circles. (D) Superpositioning of the CrIFT80 structure (coloured as in (A)) with the evolutionarily related β'COP protein (pdb: 3MKQ; coloured in grey). The two structures differ significantly with respect to the positions of β-propeller two as well as the α-solenoid C-terminus. (E) Isolated view on β-propeller 2 of β'COP (pdb:3MKQ, coloured in grey). Although this propeller also lacks the outermost strand in blade 7, it is closed by an extended loop in blade 1 (coloured in red).

DOI: https://doi.org/10.7554/eLife.33067.003

The following figure supplement is available for figure 2:

**Figure supplement 1.** Sequence alignment between IFT80 proteins from various species.
DOI: https://doi.org/10.7554/eLife.33067.004

complex with a dissociation constant (Kd) of ~225 nM (*Figure 3D*). We previously reported that IFT80 also interacts with IFT54/20 within the IFT-B2 sub-complex (*Taschner et al., 2016*). Here, we show that this interaction requires the C-terminal α-helical extension of IFT80 (*Figure 3—figure supplement 1A*). The IFT80-IFT54/20 interaction does not require the N-terminal tubulin-binding CH-domain of IFT54 (*Figure 3—figure supplement 1B*) and only very weak binding was observed between IFT80$_{1-654}$ and IFT54ΔCH/20 demonstrating that the association depends on the 111 most C-terminal residues of IFT80 (*Figure 3—figure supplement 1C*). We conclude that IFT80 is incorporated into the IFT complex mainly via a strong interaction of the BP1 domain with the IFT38 CH-domain but displays additional interactions between the IFT80 C-terminus and the IFT54/20 complex.

## IFT80 adopts a homo-dimeric structure mediated by the C-terminal domains

The structure of the αβ'-COP coatomer core complex revealed a trimer where 3 N-terminal β-propellers of β'-COP come together to form a triskelion (*Lee and Goldberg, 2010*). Given that the conserved N-terminal β-propeller (BP1) of IFT80 interacts with IFT38, such a triskelion arrangement is unlikely within the context of the IFT complex. Inspection of the crystal packing of the IFT80 structure presented here revealed an IFT80 homo-dimer organized around a crystallographic 2-fold axis (*Figure 4A*). The unusual position of the C-terminal BP2 and α-helical domains outlined above and shown in *Figure 2* allows IFT80 to homo-dimerize via an extended interaction interface contributed by BP2 and the α-helical C-terminal domain. Two separate interaction interfaces contribute to IFT80 dimerization. The first is a 6-stranded anti-parallel β-sheet where blade 7 of each BP2 domain contributes three β-strands (solid rounded rectangle in *Figure 4A*, and zoom-in in *Figure 4B*). Such an arrangement is possible in IFT80 due to the lack of the loop connecting blades 1 and 7 in related BP2s such as the one in β'COP (see *Figure 2E*). Two residues (L575 and S577) in β-strand 54 make backbone interactions with their symmetry mates (partially mediated by water molecules) to keep the resulting β-sheet aligned (*Figure 4B*). The second interface (present in two copies in the dimer due to symmetry) is formed by interactions of the first TPR-like repeat (alpha helices α4−α5) in the α-solenoid domain with blade 1 of BP2 (dashed rounded rectangles in *Figure 4A*, and zoom-in in *Figure 4C*). Only one of the contacts here is hydrophobic in nature (F605 is located in a hydrophobic pocket formed by the aliphatic parts of the side chains of R316 and E329), all other interactions are hydrophilic and partially water-mediated, and also involve several backbone interactions (*Figure 4C*). Taken together, the entire interaction interface between two IFT80 monomers (1x 'interface 1' and 2x 'interface 2') has a combined buried surface area of ~1200 Å$^2$.

## IFT80 dimerization requires the first 76 residues of the C-terminal α-helical extension

To ascertain if IFT80 is also a dimer in solution we carried out several experiments. Two important predictions about IFT80 dimerization can be drawn from our analysis of the IFT80 dimerization interfaces described above. Firstly, the dimer should be sensitive to high salt concentrations in the buffer due to the largely hydrophilic nature of inter-subunit interactions. Secondly, deletion of the α-solenoid C-terminal domain (such as in the BP1-BP2 construct) should abolish dimerization, as this destroys the larger 'interface 2', which is present twice within the IFT80 dimer (*Figure 4*). In SEC experiments, IFT80 eluted as a dimer at low salt conditions (150 mM NaCl), but as a monomer at

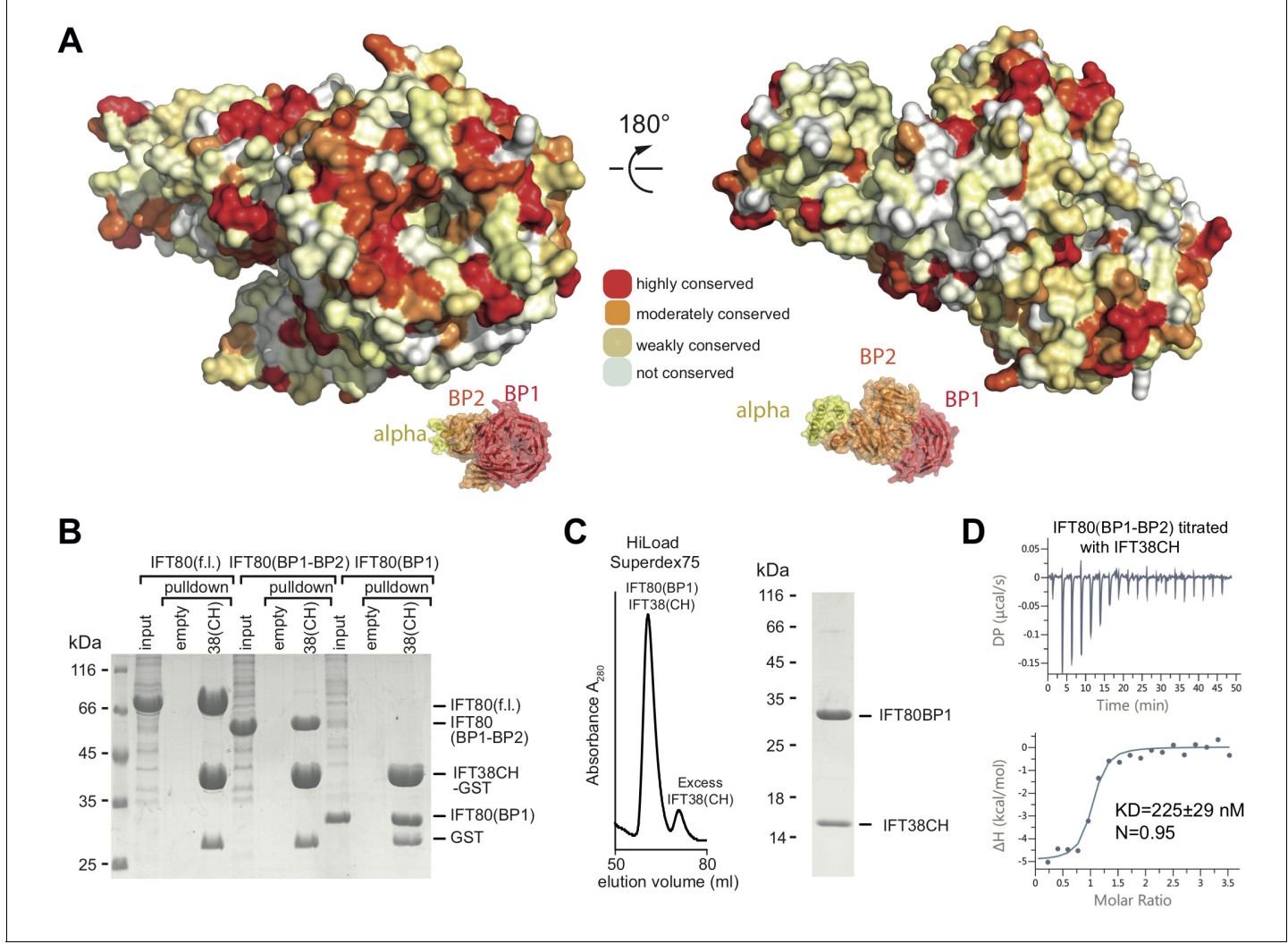

**Figure 3.** Interaction of the CrIFT38 CH domain with a conserved surface patch on CrIFT80 BP1. (**A**) Mapping of evolutionarily conserved surface-residues onto the CrIFT80 crystal structure. Two orientations (related to each other by a 180° rotation) are shown. The schematic drawing below the structure indicates the relative positions of individual domains. In the orientation on the left, several highly conserved residues (forming a pronounced patch) are shown at the top of CrIFT80 BP1. (**B**) GST-pulldown assays showing that the CrIFT80 BP1 domain is sufficient for interaction with CrIFT38CH. (**C**) The CrIFT80(BP1)/CrIFT38CH complex can be reconstituted by SEC. After mixing CrIFT80(BP1) with a slight excess of CrIFT38CH, a peak for the stable complex is obtained (left), and SDS-PAGE analysis shows that the complex contains stoichiometric levels of the two proteins (right). (**D**) Isothermal titration calorimetry (ITC) analysis of the interaction of CrIFT38CH with CrIFT80(BP1-BP2). The two proteins form a complex with a dissociation constant of 225 nM.

DOI: https://doi.org/10.7554/eLife.33067.006

The following figure supplement is available for figure 3:

**Figure supplement 1.** The C-terminus of IFT80 is required for IFT54/20 interaction.

DOI: https://doi.org/10.7554/eLife.33067.007

higher salt concentrations (500 mM NaCl, *Figure 5A*). To confirm the dimeric and monomeric states of full-length IFT80 depending on the salt concentration in the buffer, we collected small angle X-ray scattering (SAXS) data in buffers with 150 mM or 500 mM NaCl. The measurements indicated that the molecular dimensions of the IFT80 protein were larger in low salt than in high salt (*Figure 5B* and *Table 2*), again suggesting that high salt converts the IFT80 dimer into monomers. Consistent with this, the SAXS envelope obtained in low salt conditions showed a significantly better fit to the crystal structure of the IFT80 dimer ($\chi^2$ of 0.947) than to that of the monomer ($\chi^2$ of 4.419) (*Figure 5C*), whereas the opposite was the case for the data obtained in high salt ($\chi^2$ of 1.589 for

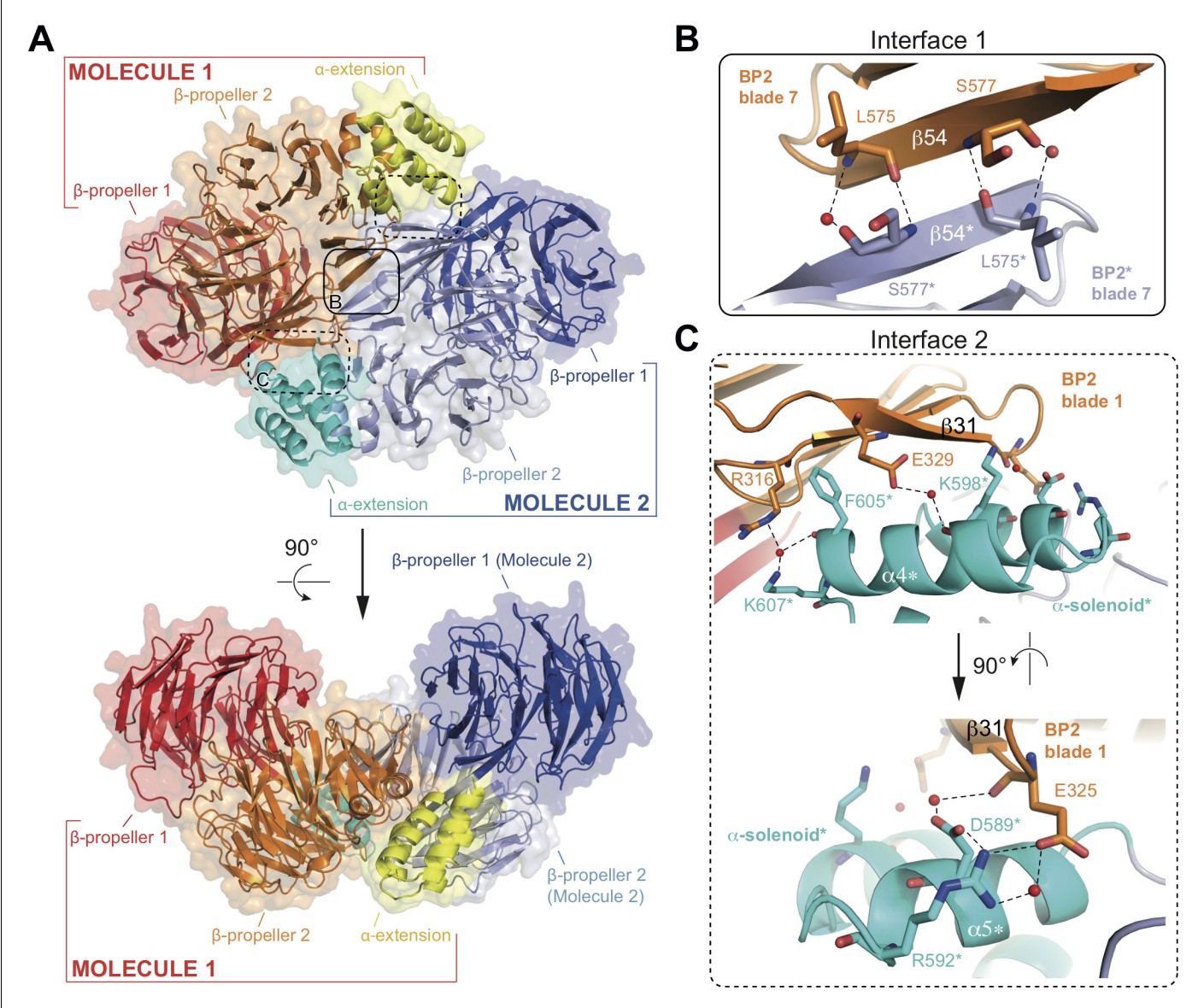

**Figure 4.** Structure of the CrIFT80 homo-dimer. (A) Two views (related by a 90° rotation) on the IFT80 homodimer in cartoon representation with transparent surface. The two molecules are coloured in red/orange/yellow and dark blue/light blue/cyan for Molecule 1 and 2, respectively. Individual domains are labelled in the same colour. Two major interfaces hold the molecules together and are indicated by boxes. The solid box indicates the location of 'interface 1' (present only once in the dimer), and the dashed boxes indicate the locations of 'interface 2' (present twice due to symmetry). Zoom-in views on the interfaces are shown in (B) and (C). (B) Zoom-in view on interface 1, formed by backbone interactions between β-strand 54 from each of the two molecules. Asterisks after the labels of domains, secondary structure elements, and residues indicate that they are found on 'Molecule 2'. The colour scheme is the same as in (A). Red spheres indicate water molecules. (C) Two perpendicular zoom-in views on interface 2, highlighting several interactions between the α-solenoid helices of Molecule two with BP2 blade 1 of Molecule 1. Asterisks after the labels of domains, secondary structure elements, and residues indicate that they are found on 'Molecule 2'. The colour scheme is the same as in (A).

DOI: https://doi.org/10.7554/eLife.33067.008

dimer, $\chi^2$ of 1.187 for monomer) (*Figure 5D*). Taken together these results confirm that IFT80 exists as a dimer in solution at physiological salt concentrations.

To determine if the C-terminal α-helical extension is required for IFT80 dimerization in solution we did SEC experiments under low salt conditions with the IFT80(BP1-BP2) construct lacking the C-terminus and the IFT80(1-654) construct containing the 126 residue α-helical segment shown to mediate dimer formation in the IFT80 crystal structure. The result show that IFT80(1-654) is a dimer

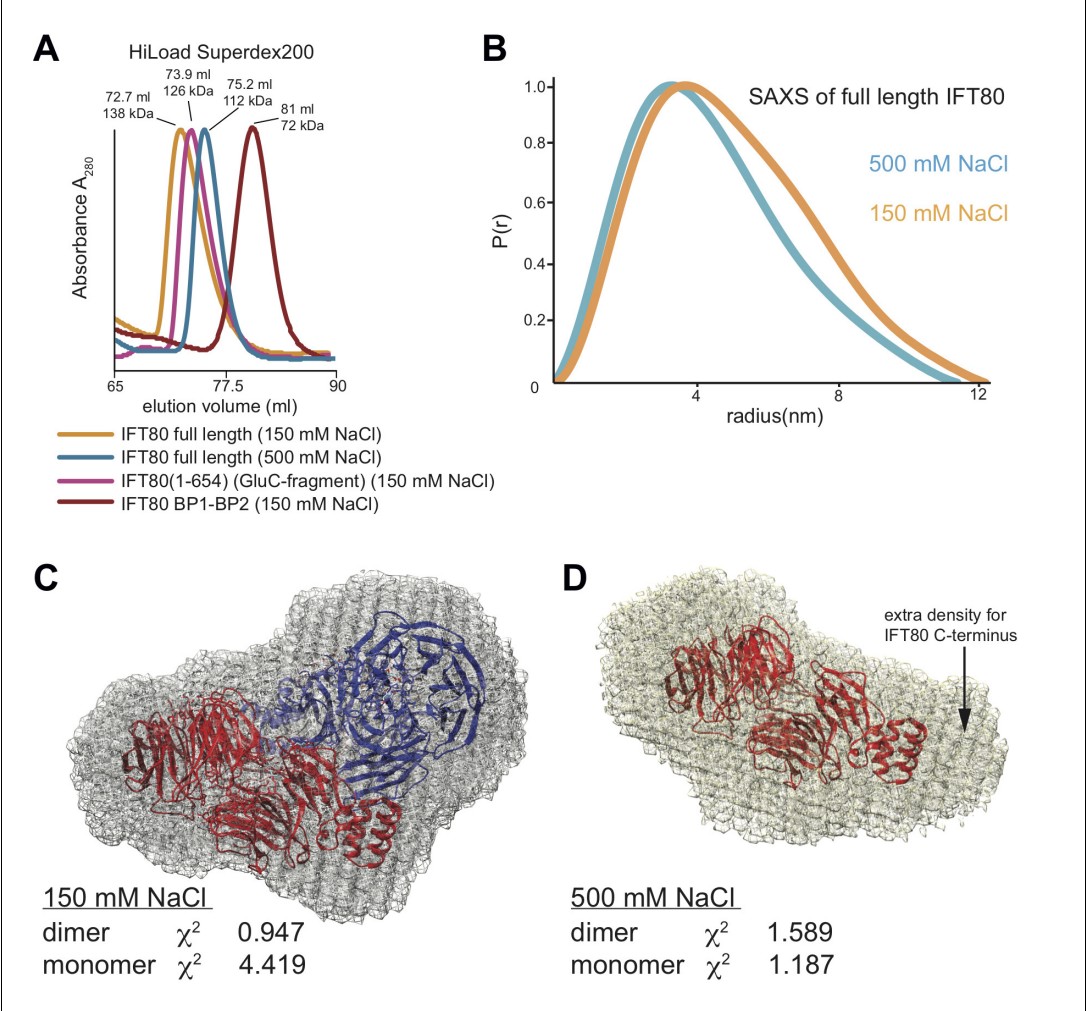

**Figure 5.** Analysis of the IFT80 homo-dimer by SEC and SAXS. (**A**) SEC analysis (HiLoad Superdex200 column) of various constructs of CrIFT80. Full-length IFT80 elutes at different positions at low and high salt concentration (150 mM and 500 mM NaCl, respectively), indicating that the homo-dimer is salt-labile. This is consistent with the fact that most interactions between the molecules are hydrophilic (see **Figure 4**). Elution volumes and apparent Mw are indicated for each peak. As SEC is shape-dependent, the Mw values will deviate from the theoretical values. All peaks are far from the void volume of 45 mL for this particular column. The proteolysed IFT80 fragment (residues 1–654) elutes between the low- and high-salt peaks of full-length IFT80, consistent with a dimer, whereas the construct lacking the entire α-solenoid domain (IFT80 BP1-BP2) elutes at significantly lower molecular weight and is monomeric even at low salt concentrations. (**B**) P(r) vs. radius plot derived from SAXS analysis of full-length IFT80 at low and high salt concentrations (150 mM vs. 500 mM NaCl, respectively), indicating that an IFT80 dimer at low salt is broken up into monomers at high salt. (**C**) The SAXS envelope calculated for IFT80 at low salt concentrations (grey mesh) provides a much better fit to the crystal structure of the IFT80 homo-dimer (individual molecules shown in cartoon representation in red and blue, respectively), providing proof that IFT80 exists as a homo-dimer in solution at 150 mM NaCl. (**D**) The SAXS envelope calculated for IFT80 at high salt concentrations (grey mesh) fits better to the crystal structure of a monomeric form of IFT80 (shown in cartoon representation in red), indicating the dimer is broken up into monomers at 500 mM NaCl. Extra density in the SAXS envelope likely corresponds to the C-terminus of IFT80, which is not present in the crystallographic model.

DOI: https://doi.org/10.7554/eLife.33067.009

whereas IFT80(BP1-BP2) is a monomer (**Figure 5A**). To further characterize the IFT80 monomer-dimer equilibrium, we employed dynamic light scattering (DLS) measurements in addition to SAXS, using a range of concentrations of purified IFT80 protein. Consistent with the assumption that the BP1-BP2 construct of IFT80 is monomeric, DLS measurements resulted in a constant ~3.6 nm radius of the molecule independently of protein concentration (**Figure 6A**, left). This value agrees with the dimension of the BP1-BP2 region obtained from the IFT80 crystal structure. In contrast, the obtained radius for the full-length protein changed with increasing protein concentration, starting just below 4 nm at the lowest concentration examined, and reaching a plateau around ~5.5 nm when the

**Table 2.** Small-angle X-ray scattering (SAXS) data

**Data-collection parameters**

| Instrument: | ESRF BM29 | | | |
|---|---|---|---|---|
| Wavelength (Å) | 0.99 | | | |
| q-range (Å$^{-1}$) | 0.0032–0.49 | | | |
| Sample-to-detector distance | 2.867 m | | | |
| Exposure time (sec/frame) | 1 | | | |
| Temperature (K) | 283 | | | |
| Detector | Pilatus 1M (Dectris) | | | |
| Flux (photons/s) | $1 \times 10^{12}$ | | | |
| Beam size (μm$^2$) | 172 × 172 | | | |
| **Structural parameters** | IFT80 | | IFT80(BP1-BP2) | IFT80 |
| Type of experiment | SEC SAXS | SEC SAXS | Concentration series | Concentration series |
| Concentration used (mg/mL) | 2 | 2 | 0.47–7.5 | 0.5–8.0 |
| NaCl concentration (mM) | 150 | 500 | 150 | 150 |
| From* $R_g$ (Å) | 38.7 | 34.7 | 26.2 | 41.4–55.7 |
| p(r)* Dmax (Å) | 122 | 114 | 86.1 | 140.4–194.4 |
| Porod volume Vp x10$^3$ (Å3) | 210.8 | 119.8 | 94.3 | 216–347 |
| Molecular mass (kDa) from Vp | 131.8 | 74.8 | 58.9 | 126–202 |
| Molecular mass (kDa) from sequence | 170 (dimer) | 85 (monomer) | 65 | 85 |
| **Modeling** | | | | |
| Dammif NSD† | 0.597 | 0.795 | | |

*distance distribution function

†normalized spatial discrepancy between the 10 calculated models

DOI: https://doi.org/10.7554/eLife.33067.010

concentration was increased (*Figure 6A*, right). Again, these dimensions agree with monomeric and dimeric forms of IFT80 obtained from the crystal structure. Similar results for both BP1-BP2 and fl IFT80 were obtained using SAXS (*Figure 6B*), with the pair-distance distribution function (P(r)) revealing a concentration-dependent increase in size for full-length IFT80 but not for the IFT80 (BP1-BP2) construct. The results shown in Figures 4-6 demonstrate that IFT80 dimerization requires the first 72 residues but not the most C-terminal 111 residue of the α-helical extension.

## Concentration dependent multimerization of a reconstituted 14-subunit IFT-B complex

To address if dimerization might also occur in an IFT80-containing IFT-B complex, we assembled a 14-subunit IFT-B complex containing a 9-subunit IFT-B1 complex (IFT88/71/74/70/52/46/27/25/22) (*Taschner et al., 2014*) and a 5-subunit IFT-B2 complex (IFT80/57/54/38/20) (*Figure 6C*). This complex is similar to a previously reconstituted 15-subunit assembly (*Taschner et al., 2016*), but is missing IFT172. The IFT172 protein, which was omitted from the complex, binds more weakly to the IFT-B complex than the other subunits and has a tendency to dissociate (*Taschner et al., 2016*). Both DLS (*Figure 6D*) and SAXS measurements (*Figure 6E*) over a range of concentrations of this 14-meric IFT-B complex revealed results paralleling those of fl IFT80 alone, with a concentration-dependent increase in the radius of the complex from a starting value of ~10 nm to a plateau at a value of approximately 15 nm. Due to the lack of suitable high-resolution structures of the IFT-B complex, we can at this point not compare these dimensions to measurements from the structure. The values are, however, compatible with a concentration dependent transition from monomeric to dimeric 14-subunit IFT-B complexes. The fact that the concentration-dependent behaviour of the 14-subunit IFT-B complex is similar to isolated IFT80 suggests that IFT80 is capable of organizing dimers

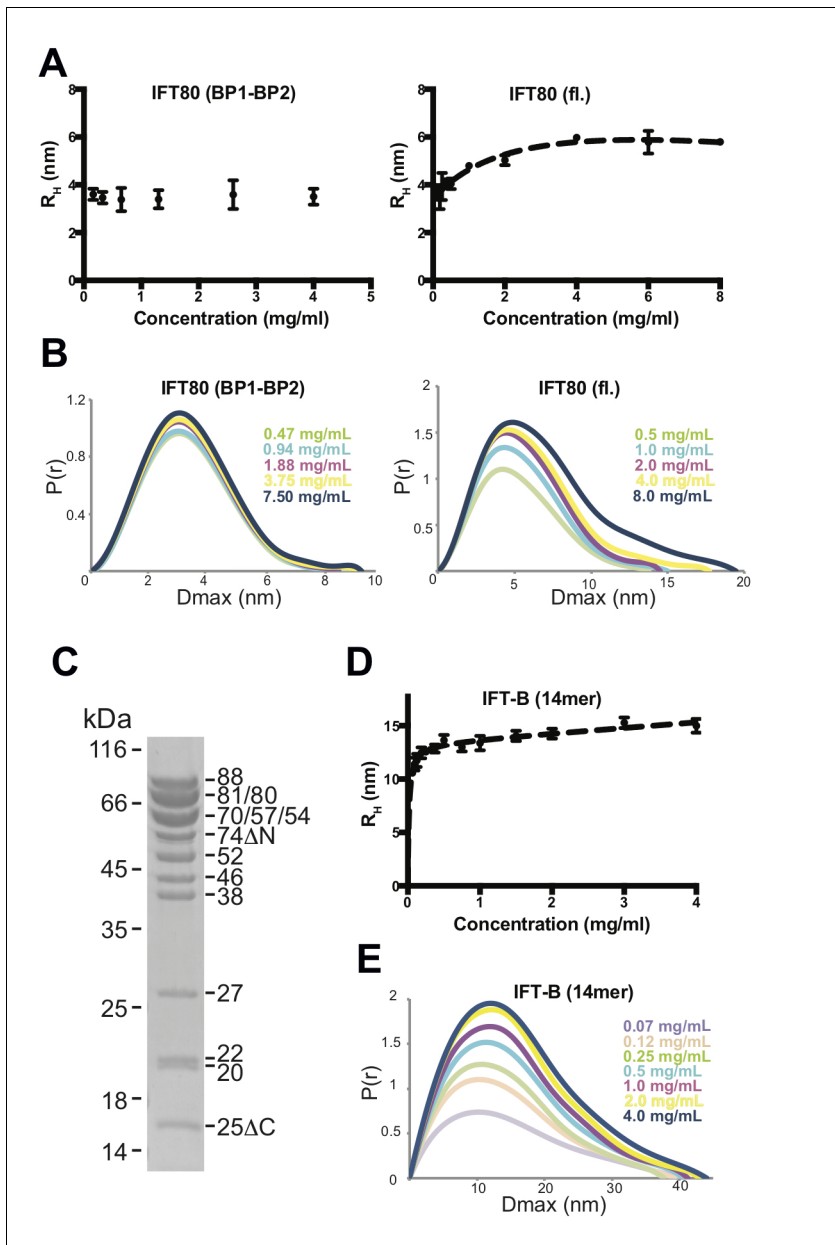

**Figure 6.** Analysis of IFT80 and the IFT-B complex by DLS and SAXS. (**A**) (left) DLS analysis of the IFT80 (BP1-BP2) construct shows that the hydrodynamic radius ($R_H$) is consistent with a monomeric form of the protein, and does not change with increasing concentration. (right) DLS analysis of full-length IFT80 indicates that the radius increases at higher concentrations, consistent with a concentration-dependent IFT80 homo-dimerization. (**B**) SAXS analysis (P(r) vs. Dmax plots indicates that whereas the Dmax of full-length IFT80 increases with increasing concentration (right), this is not the case for the truncated IFT80 (BP1-BP2) construct (left), confirming the results obtained by DLS shown in (**A**). (**C**) Coomassie-stained SDS-gel showing the purity of the reconstituted 14-subunit IFT-B complex used for the experiments shown in (**D**) and (**E**). (**D**) DLS analysis of a 14-subunit IFT-B complex containing IFT80 shows a concentration-dependent increase of the hydrodynamic radius, indicating IFT-B dimerization. (**E**) SAXS analysis (P(r) vs. Dmax plot) also indicates IFT-B dimerization at increasing protein concentrations, paralleling the DLS result shown in (**D**).

DOI: https://doi.org/10.7554/eLife.33067.011

of IFT-B complexes consistent with the multimerization of IFT particles in to train-like structures observed in situ (*Kozminski et al., 1993*; *Pigino et al., 2009*).

## IFT80 is absolutely required for initiation of ciliary axoneme assembly

Assembly of the ciliary axoneme involves recruitment of α/β-tubulin heterodimers to the transition zone of cilia, possibly piggybacking on cytoplasmic vesicles, and their subsequent transport to the ciliary tip by IFT trains (*Wood and Rosenbaum, 2014*; *Craft et al., 2015*; *Hao et al., 2011*; *Kubo et al., 2016*). The IFT-B1 subcomplex is involved in the recruitment of α/β-tubulin heterodimers through direct interaction with IFT81/74 (*Bhogaraju et al., 2013*). In previous studies of IFT-B2 subcomplex components, including IFT80, a significant proportion of cells were able to produce acetylated tubulin-positive cilia following gene knockdown (*Absalon et al., 2008*; *Beales et al., 2007*; *Yuan and Yang, 2015*; *Yuan et al., 2016*; *Yang and Wang, 2012*; *Wang et al., 2013*; *Rix et al., 2011*). It is therefore not clear whether IFT-B2 components are dispensable for IFT complex/train formation or α/β-tubulin recruitment, or whether this reflects incomplete gene inactivation.

We used gene-editing to achieve total gene knockout by generating clonal IMCD3 cell lines carrying biallelic frameshift mutations in mouse *Ift80* (*Figure 7—figure supplement 1A*). Using a gRNA targeting exon 3, we isolated one clone that was compound heterozygous for p.T24fsX32/p.V20fsX29 mutations, as judged by direct sequencing and T7 endonuclease I assay (*Figure 7—figure supplement 1B*). Both alleles are predicted to cause premature termination at the start of BP1. Cilia formation was monitored by immunostaining to label the ciliary axoneme (acetylated tubulin) and membrane (Arl13b), and the basal body (γ-tubulin). Serum-starvation of control wild-type cells for 48 hr resulted in robust formation of primary cilia. By contrast, cilia completely failed to form in the *Ift80* mutant cell line – 100% of mutant cells displayed absolutely no acetylated tubulin staining, while a single γ-tubulin + basal body could be seen overlaid by a small amount of Arl13b + ciliary membrane (*Figures 7A* and *8A*).

To control for off-target mutations and clonal artefacts, we also characterised two further clonal cell lines. One clone appeared homozygous for a p.T23fsX31 mutation also in exon 3. Another clone generated using an independent gRNA in exon 9 of *Ift80* was compound heterozygous for p.C288fsX309/p.G289fsX309 mutations (*Figure 7—figure supplement 1C–D*). These alleles predict termination at the beginning of BP2, and qRT-PCR showed that they cause transcript instability (*Figure 7—figure supplement 2*). Both of these clones also completely lacked cilia (*Figure 7—figure supplement 3*). We confirmed that these *Ift80* knockout phenotypes were not secondary to defects in cell cycle dynamics or DNA damage repair, which have been linked to defective ciliogenesis (*Figure 7—figure supplements 4–5*). We conclude that IFT80 is absolutely required to form cilia.

## Deletion of the IFT80 dimerization domain inactivates protein function, but missense mutations do not

In order to visualise IFT80 within the ciliary axoneme, we created a mouse IFT80-GFP (mIFT80-GFP) C-terminally tagged expression construct that expressed efficiently in IMCD3 cells. Using super-resolution fluorescence microscopy, we observed mIFT80-GFP (stained using an anti-GFP antibody) within IFT trains, which punctuated the length of the cilium (*Figure 7B*). There was an average of 21.8 ± 9.6 (95% CI) particles per cilium separated by 0.33 ± 0.15 µm (95% CI). This tagged form of IFT80 was functional, rescuing ciliogenesis in *Ift80* null-mutant cells to the same degree as an untagged construct (*Figures 7C* and *8*).

Currently, six missense mutations in *IFT80* have been identified as causative in eight individuals with JATD, each of which is private to individual families. No clear differences in clinical severity are noted between patients either between or within families. All residues affected by these mutations are completely conserved in vertebrates and well conserved in *C. reinhardtii*. Using the crystal structure of *Cr*IFT80, we mapped the structural position and interactions of the reported patient mutations (*Figure 8—figure supplement 1*). p.H105Q, p.L143S and p.G241R mutations reside within BP1 (*Figure 8—figure supplement 1A–C*), whereas the p.A701P and p.R719C mutations are located within the very C-terminal α-extension required for IFT54/20 binding (not observed in our crystal structure). Interestingly, the p.L549del mutation is located at the very end of an α-helical extension found between β-strands β50 and β51 of BP2 (*Figure 8—figure supplement 1D*). Therefore, *IFT80*

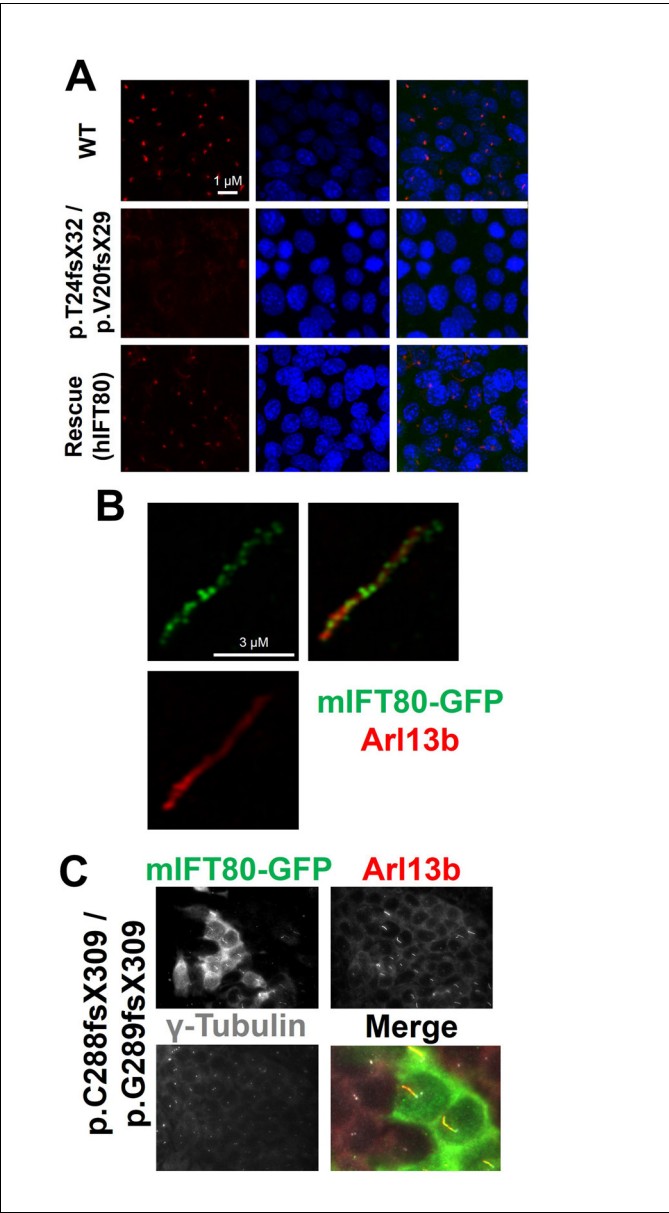

**Figure 7.** IFT80 is absolutely required for ciliary axoneme assembly. (**A**) Complete absence of acetylated tubulin staining in IMCD3 cells carrying biallelic *Ift80* frameshift mutations is rescued following transfection of human IFT80. (**B**) Super-resolution microscopy showing mIFT80-GFP IFT trains within cilia, co-localising with ciliary membrane marker ARl13b. (**C**) For the p.C288fs/pG289fs mutant clone, transfection of mouse IFT80-GFP is shown. Note that only GFP +cells produce cilia, while adjacent GFP- cells have a basal body but no cilia, demonstrating a cell autonomous requirement for IFT80.

DOI: https://doi.org/10.7554/eLife.33067.012

The following figure supplements are available for figure 7:

**Figure supplement 1.** Characterization of *Ift80* mutant cell clones.
DOI: https://doi.org/10.7554/eLife.33067.013

**Figure supplement 2.** Analysis of WT and mutant cell clones by qRT-PCR.
DOI: https://doi.org/10.7554/eLife.33067.014

**Figure supplement 3.** Analysis of ciliogenesis in multiple mutant cell clones.
DOI: https://doi.org/10.7554/eLife.33067.015

**Figure supplement 4.** Quantification of DNA damage by H2AX staining.
DOI: https://doi.org/10.7554/eLife.33067.016

**Figure supplement 5.** Analysis of cell cycle.

*Figure 7 continued on next page*

*Figure 7 continued*

DOI: https://doi.org/10.7554/eLife.33067.017

missense mutations are distributed across different functional domains of the protein, and are unlikely to affect dimerization.

We next tested the ability of missense mutant forms of mIFT80-GFP to rescue ciliogenesis. All point mutant forms of mIFT80-GFP rescued cilia formation to the same degree as wild-type (*Figure 8B–D*). We also found that all missense mutant forms of mIFT80-GFP localised to puncta within the ciliary axoneme, and quantification of the proportion of GFP +cilia revealed no difference to the wild-type (*Figure 8B and D* and *Figure 8—figure supplement 2*). Western blotting showed that all transfected missense mutant forms of mIFT80-GFP were expressed at significantly lower levels than wild-type, suggesting that these mutations cause protein instability (*Figure 8—figure supplement 3*). Collectively, these data confirm that point mutations in *IFT80* found in JATD retain a significant degree of protein function with respect to ciliogenesis and IFT.

To investigate if IFT80 dimerization is functionally relevant we carried out rescue experiments using deletion constructs all expressing at similar or higher levels than wild-type (*Figure 3—figure supplement 1D–E*). GFP-tagged human wild-type IFT80 rescues the ciliogenesis phenotype demonstrating that the GFP tag does not impair IFT complex formation and that the human construct is a suitable replacement for mouse Ift80 (*Figure 9*). Point mutations of IFT80 dimerization interface residues did not break up the homo-dimer, presumably because of the highly extended interface containing many sequence-unspecific backbone-backbone interactions (data not shown). The $IFT80_{1-657}$ construct that lacks the IFT54/20 binding domain but contains the IFT80 dimerization domain provides a partial rescue with about half the amount of cells forming cilia compared to the full-length IFT80 rescue (*Figure 9B–C*). The reduction of cilium formation in the $IFT80_{1-657}$ rescue experiment is likely a result of the reduced ability of this construct to bind IFT54/20 (*Figure 3—figure supplement 1C*). However, given that IFT80 is absolutely required for cilium formation in these cells, the partial rescue demonstrates that the IFT54/20 binding domain of IFT80 is not mandatory for initiation of ciliogenesis or IFT. In contrast, cells expressing IFT80 constructs containing only BP1 ($IFT80_{1-300}$) or BP1-BP2 ($IFT80_{1-586}$) are completely unable to ciliate (*Figure 9B–C*), demonstrating that the C-terminal α-helical domain of IFT80 is essential for cilium formation. Both $IFT80_{1-300}$ an $IFT80_{1-586}$ constructs contain the BP1 domain that links IFT80 to the IFT complex via IFT38 (*Figure 3*). The inability of $IFT80_{1-586}$ to rescue ciliogenesis is thus not a direct result of IFT80 not associating with the IFT complex. As $IFT80_{1-586}$ fails to rescues ciliogenesis and $IFT80_{1-657}$ provides a partial rescue of ciliogenesis (*Figure 9B–C*), residues 587–657, which are required for IFT80 dimerization (*Figure 6*), thus appear to be essential for cilium formation. Taken together, these data suggest that IFT80 homodimerization is required for cilium formation in mammalian cells.

## Discussion

EM tomograms of IFT trains in situ revealed doublet rows of IFT particles that multimerize along the axis of the cilium (*Pigino et al., 2009*) although the molecular basis of this multimerization is unknown. The dimerization of IFT80 observed in the crystal structure presented here has 2-fold symmetry and could thus constitute the molecular basis of IFT particle doublet formation. The IFT80 dimerization is, however, unlikely to form the basis for train formation along the axonemal axis, as this relies on an asymmetric interface (*Pigino et al., 2009*) (*Figure 10*). This notion agrees with the observation that reconstituted IFT-B complexes containing IFT80 do not appear to form very long 'train-like' oligomers (*Figure 6C–E*) (*Taschner et al., 2016*). Currently, the mechanisms and proteins involved in IFT particle oligomerization into trains along the axonemal axis are not known. It is tempting to speculate that this oligomerisation could be the result of association of IFT-B with factors such as the IFT-A complex or IFT motors.

How might an IFT80 dimer be accommodated in the IFT-B complex? One possibility is that each fully assembled IFT-B complex contains two copies of IFT80, but only one copy of the other IFT-B proteins. However, this is unlikely for two reasons. Firstly, judging from previous reconstitutions of both a 6-subunit IFT-B2 complex as well as of a 15-subunit IFT-B (B1 and B2) complex, these assemblies appear to contain stoichiometric amounts of all subunits, including IFT80 (*Taschner et al.,*

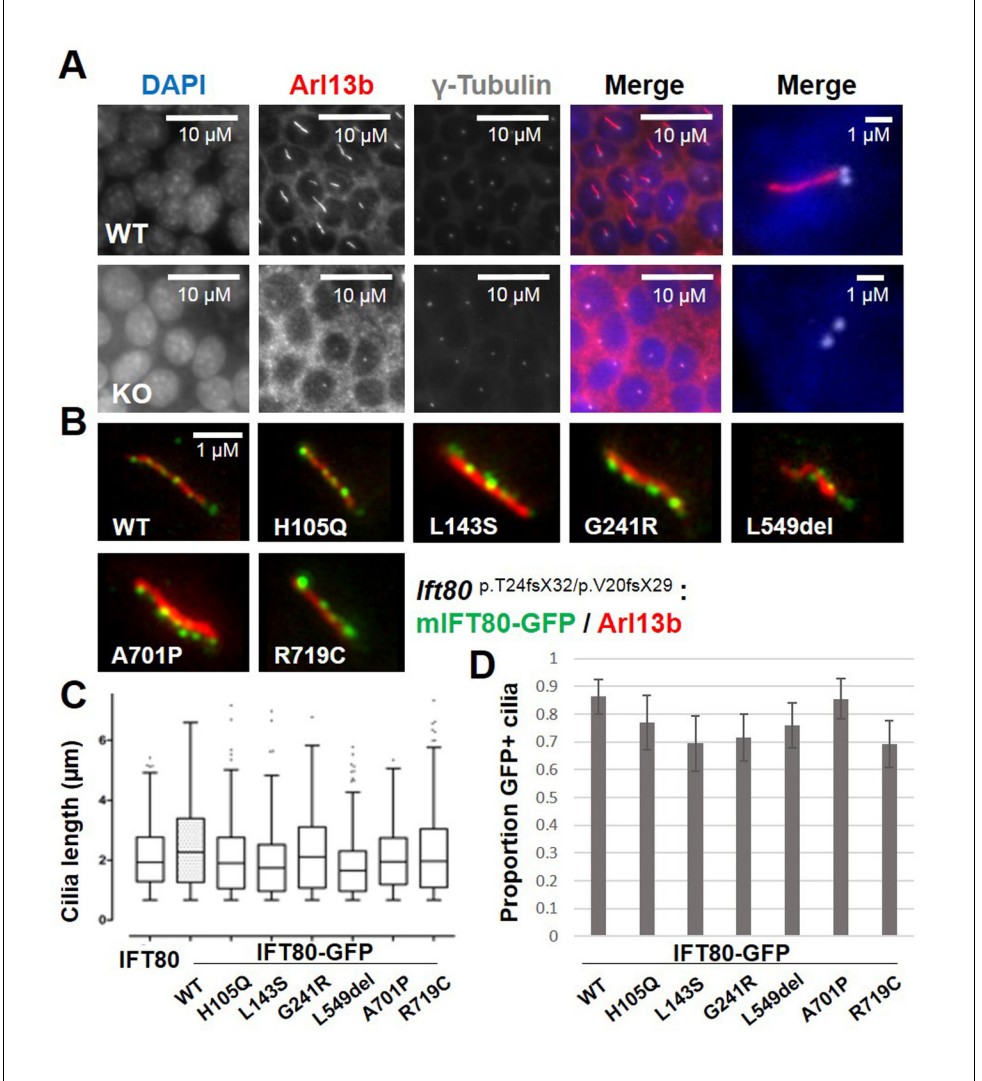

**Figure 8.** Functional analysis of IFT80 missense mutations. (A) Immunoflourescence analysis of *Ift80* KO cells showing that 100% of cells exhibit reduced ciliary membrane formation (Arl13b staining), while basal body formation (γ-tubulin) is unaffected. (B) Failed ciliogenesis in *Ift80* KO cells is fully rescued following transfection of WT and missense mutant forms of IFT80-GFP. Punctate staining for IFT80-GFP is observed for all variants, representing IFT trains. (C) Quantification of cilia length (Arl13b) shows that transfection of untagged WT mIFT80 rescues failed ciliogenesis to the same degree as mIFT80-GFP. All mutants rescued ciliogenesis to the same degree as WT (n = 3 biological replicates; error bars are standard deviations). (D) Quantification of the proportion of GFP +cilia co-localizing with Arl13b staining following transfection of different mIFT80-GFP point mutations. (n = 3 biological replicates; error bars are standard errors of the mean; t-test).
DOI: https://doi.org/10.7554/eLife.33067.018

The following figure supplements are available for figure 8:

**Figure supplement 1.** Structural analysis of IFT80 residues mutated in human ciliopathy patients.
DOI: https://doi.org/10.7554/eLife.33067.019

**Figure supplement 2.** Quantification of functional rescue experiments in a second independent *Ift80* mutant cell clone.
DOI: https://doi.org/10.7554/eLife.33067.020

**Figure supplement 3.** Western blot analysis of wild-type and missense mutant forms of mIFT80-GFP.
DOI: https://doi.org/10.7554/eLife.33067.021

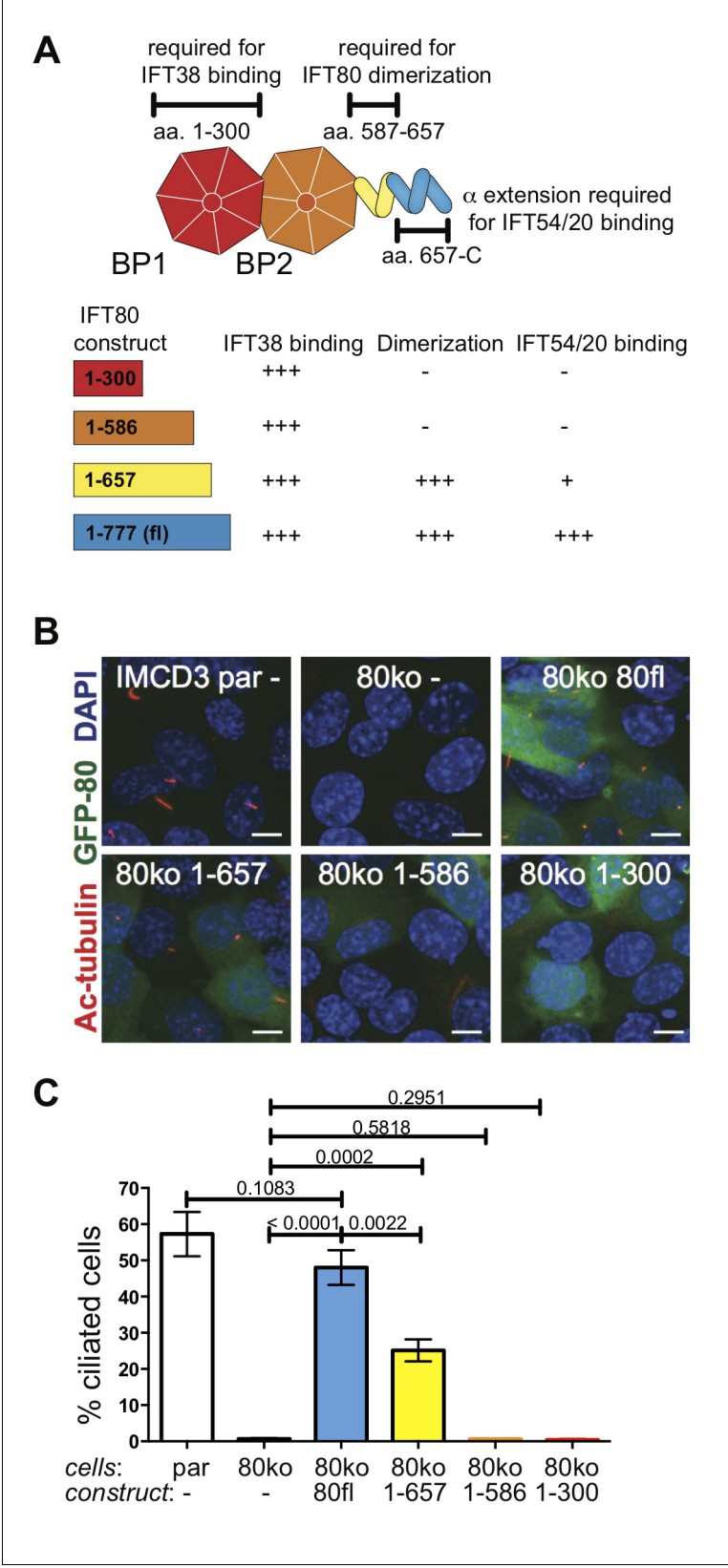

**Figure 9.** IFT80 dimerization domain is required for proper ciliogenesis in IMCD3 cells. (**A**) Schematic of the human IFT80 protein architecture with the various functions of domains assigned according to the data presented in *Figures 3–6*. (**B**) Representative fluorescent microscopy images of parental (par) IMCD3 cells and IMCD3 cells

*Figure 9 continued on next page*

*Figure 9 continued*
with IFT80 knocked out by CRISPR/Cas (80ko) and rescue experiments with various human IFT80 constructs tagged with GFP (green) stained for cilia with acetylated tubulin antibody (red) and for nuclei with DAPI (blue). Scale bars represent 10 μm. (C) Quantification of the IMCD3 ciliogenesis rescue experiment shown in B (n = 3, biological replicates with at least 400 cells analyzed from four random areas, mean ±SD, unpaired t-tests).
DOI: https://doi.org/10.7554/eLife.33067.022

*2016*). Secondly, the ITC data displayed in *Figure 3D* reveal a 1:1 IFT80:IFT38 complex. The second possibility is that only 1 copy of IFT80 is present per IFT-B complex, but that the entire IFT-B complex can dimerize via the IFT80 subunit. This possibility is compatible with the IFT80 dimer shown in *Figure 4* as the most N-terminal β-propeller is available to interact with IFT38 thus allowing incorporation of an IFT80 dimer into the IFT complex. However, the exact molecular basis of IFT train formation will have to await the structural elucidation of higher order IFT complexes by X-ray crystallography, single particle EM or high-resolution EM tomographic reconstructions of isolated or reconstituted IFT trains.

The crystal structure of IFT80 presented here, together with our characterisation of distinct interaction domains, provides a framework to consider the functional effects of disease causing missense mutations. These mutations are distributed throughout the protein, occurring in distinct functional

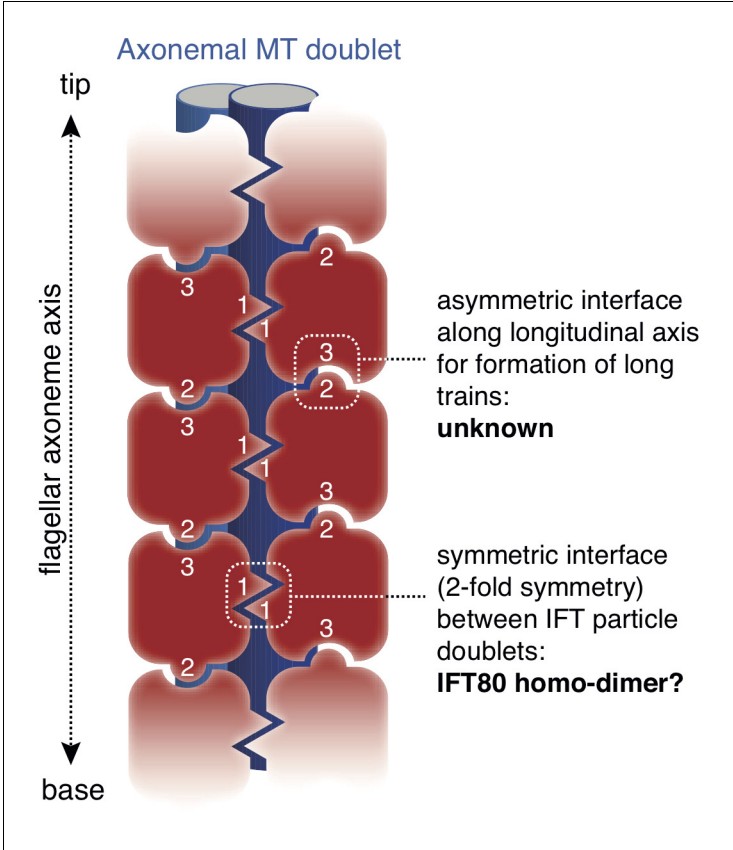

**Figure 10.** Model for the putative role of IFT80 the formation of IFT trains. Electron tomography reconstructions of IFT trains in situ in *Chlamydomonas reinhardtii* flagella showed doublets of IFT particles repeated in a head-to-tail fashion along the longitudinal axis of the flagellar axoneme (*Pigino et al., 2009*). The 2-fold symmetry at the interface between particle doublets might be explained by homo-dimerization of the IFT complex through IFT80 (indicated with number '1'). However, IFT80 homo-dimerization cannot explain the head-to-tail arrangement of IFT particles, as this relies on an asymmetric interface (indicated with numbers '2' and '3').
DOI: https://doi.org/10.7554/eLife.33067.023

domains of the protein. All mutant forms of IFT80 contributed to IFT trains within cilia and, unlike a deletion mutant lacking the dimerization domain, rescued ciliogenesis in *Ift80* null-mutant cells to the same degree as wild-type protein. Taken together, these observations suggest that missense mutations do not ablate IFT-B complex interactions or homodimerization of the protein. Given the genetic threshold relationship between IFT80 dosage and ciliogenesis (*Rix et al., 2011*), it is more likely that missense mutations cause partial protein instability to a degree that is consistent with recruitment of cargo to cilia and IFT. In the future, it will be necessary to test this hypothesis in missense mutant mice.

## Materials and methods

### Cloning and purification of IFT protein constructs used in this work

Full-length CrIFT80 (residues 1–765), and shorter C-terminal truncations (BP1-BP2, residues 1–582; and BP1; residues 1–299), were amplified from *Chlamydomonas reinhardtii* cDNA with a forward primer containing an EcoRI restriction site, and a reverse primer containing six additional histidine codons and an XbaI restriction site. The resulting PCR products were cloned between the EcoRI and XbaI sites in MCS2 of the vector pFL (*Bieniossek et al., 2008*). The resulting plasmid was transformed into DH10Bac competent *E.coli* cells (Thermo Fisher Scientific, Waltham, MA). Recombinant baculoviral DNA was purified, and baculoviruses produced in the Sf21 cell line (Thermo Fisher Scientific) as described previously (*Taschner et al., 2016*). After titration of the virus, 3–6 liters of HighFive cells (Thermo Fisher Scientific) were infected at a cell density of $10^6$ cells/ml for routine expression of the various constructs. Cells were harvested 72 hr post-infection by centrifugation (1000 g, 15 min), and cytosolic extracts were prepared as described (*Taschner et al., 2016*). Lysates were first loaded on a $Ni^{2+}$-NTA column (5 ml, Sigma-Aldrich, St. Louis, MO), and after extensive washes (20 cV) with high-salt wash buffer (50 mM Tris-HCl pH7.5, 1 M NaCl, 10% glycerol) the bound material was eluted with Ni-elution buffer (20 mM Tris-HCl pH7.5, 50 mM NaCl, 500 mM Imidazole and 10% glycerol). The eluate was immediately loaded onto a Q-Sepharose column (5 ml, GE Healthcare, Chicago, IL), and the bound proteins eluted in a 70 mL NaCl gradient (50–500 mM). Fractions containing the IFT80 protein were concentrated, and further separated on size-exclusion chromatography (SEC; HiLoad200 column) in IFT80 SEC buffer (20 mM Tris-HCl pH8.0, 200 mM NaCl, 5% glycerol) to remove remaining contaminants. Fractions containing pure material were concentrated and snap-frozen in liquid nitrogen.

Other proteins used in this study were produced as described previously (*Taschner et al., 2016*). For the reconstitution of the IFT80(BP1)/IFT38CH complex shown in *Figure 3C*, purified IFT80(BP1) protein was mixed with a 10% molar excess of purified IFT38CH, incubated at 4°C for 1 hr and then subjected to SEC (HiLoad Superdex75) in IFT80 SEC buffer.

We used gateway cloning to insert full-length mouse Ift80 into pHIV-GFP, pDEST47 or pDEST53, to create untagged, C- or N-terminally tagged Ift80.

### Subculture of IMCD3 cells

mIMCD-3 cells (ATCC CRL-2123) cultured in DMEM/F12 medium (Thermo Fisher Scientific) supplemented with 10% FBS, at 37°C in 95% air and 5% CO2. For subculture of cells in 75 cm2 flasks, monolayers were washed in 5 ml Dulbecco's phosphate-buffered saline (DPBS). Cells were detached by incubation with 2 ml 0.25% (w/v) trypsin-EDTA at 37°C for 5–15 min. Cells were dispersed in 6 ml medium and pelleted by 5 min centrifugation at 200 g. Trypsin-containing medium was aspirated and cells re-suspended in fresh medium and seeded in a new 75 cm2 flask in a 1:10 ratio. Reagent volumes were scaled for subculture in vessels of varying sizes.

### CRISPR/Cas9 generation of mutations in Ift80

gRNAs were ordered as pairs of 5'-phosphorylated oligos. At a concentration of 100 µM, oligos were annealed in a thermocycler: 37°C, 30 min; 95°C, 5 min; ramp down to 25°C at −5 °C/min. 100 ng circular plasmid pX330 (Addgene ID 42230) was digested with 4.0 µl BbsI (5000 U/ml) restriction enzyme (NEB, Ipswich, MA) in a 50 µl reaction with 1X NEBuffer 2.1 (NEB), for 1 hr at 37°C. To make respective CRISPR expression vectors, gRNAs annealed oligos were ligated in linearised pX330. Ligations were carried out in 10 µl reactions in 1X T4 DNA ligase buffer (NEB), using 50 ng linear

pX330, 1 µl annealed oligos (0.5 µM) and 1 µl T4 DNA ligase (NEB), and were incubated at 4°C for 16 hr. Correct sgRNA insertion was confirmed by sequencing. Expression vectors were amplified and purified by Plasmid Midi Kit (QIAGEN, Germany).

In a well of a six well plate, 40–60% confluent IMCD3 cells were transfected with 2.4 µg plasmid DNA, using 8.5 µl Lipofectamine 2000 transfection reagent (Thermo Fisher Scientific) in 300 µl Opti-MEM reduced serum medium (Thermo Fisher Scientific). 2.4 µg plasmid DNA was made up in equal parts of each expression vector (0.8 µg each vector for triple transfections; 1.2 µg each vector for co-transfections). For positive control of transfection, cells were transfected with the GFP expression vector alone (0.8 µg plasmid when controlling for triple transfections; 1.2 µg when controlling for co-transfections). For negative control, cells were mock-transfected with transfection reagent and sterile DPBS, but no plasmid DNA.

GFP +cells were selected by FACS. A population of GFP +cells was collected from the group sort. Single cell sorted cells were seeded in 96 well plates in 150 µl conditioned medium. Conditioned medium was prepared by sterile filtration of a 1:1 mixture of fresh medium, and medium removed from a flask of cells in exponential growth phase (50–70% confluent). Single cell sorted colonies were expanded by subculture to increasingly larger culture vessels (24 well plate, 12 well plate, T25 flask).

## DNA damage assay

Sterilised round glass coverslips (13 mm diameter, no. 0; VWR) were placed in 12 well plates, one per well. $1.0 \times 10^5$ cells were seeded per well in 2 ml DMEM/F12 medium supplemented with 10% FBS. Assays were performed 24 hr after seeding, once cells had reached ~40% confluence. Cells were incubated in fresh complete medium containing 2 mM hydroxyurea (Sigma-Aldrich) for 2 hr. Untreated cells were incubated in fresh complete medium for 12 hr. Cells on coverslips were washed three times in DPBS prior to fixation in ice-cold 100% methanol for 10 min at −20°C. Cells were washed three times in DPBS and blocked in 10% goat serum in DPBS for 1 hr. Coverslips were incubated with mouse monoclonal anti-phospho-histone H2A.X (Ser139), clone JBW301, 1:800 (05–636, Merck Millipore, Burlington, MA) for 16 hr at 4°C. Cells on coverslips were washed three times in DPBS with gentle shaking, before incubation with goat anti-mouse IgG1 Alexa Fluor 488, 1:500 (997812, Invitrogen/Thermo Fisher Scientific) for 1 hr at room temperature. Secondary antibody was diluted in 10% goat serum in DPBS. Cells on coverslips were washed three times in DPBS with gentle shaking. Coverslips were mounted on glass slides in Fluoroshield with DAPI (GeneTex, Taiwan), and edges sealed with clear varnish. three images per coverslip were taken on a widefield fluorescence microscope, using a 40x objective. Identical microscope settings, including exposure duration, were used for all images. Images were analysed using FIJI (*Schindelin et al., 2012*). Nuclei were identified and segmented using the DAPI stain, and intensity of the γH2AX stain within each nucleus was measured.

## Pi-facs

Cells were cultured to 40% confluence in a 25 cm$^2$ flask and washed in 5 ml Dulbecco's phosphate-buffered saline (DPBS). Cells were detached by incubation with 0.5 ml 0.25% (w/v) trypsin-EDTA at 37°C for 5–15 min. Cells were dispersed in 3 ml medium and pelleted by 5 min centrifugation at 200 g. Trypsin-containing medium was aspirated and the cell pellet washed twice by resuspension in DPBS. Cells were again pelleted by 5 min centrifugation at 200 g, and DPBS aspirated. Cells were fixed in 1 ml cold 70% ethanol, added dropwise to the cell pellet whilst vortexing. Cells were incubated on ice for 30 min. Prior to analysis by flow cytometry, cells were pelleted by 10 min centrifugation at 1000 g, and supernatant discarded. Cell pellets were twice washed by resuspension in DPBS, and pelleted again by 10 min centrifugation at 1000 g. Cell pellets were treated directly with 50 µl RNase A solution (100 µg/ml in DPBS) to prevent staining of RNA by propidium iodide (PI). DNA was stained by addition of 400 µl PI (50 µg/ml in DPBS) directly to cells in RNase A solution, and incubation for 10 min at room temperature. Samples were analysed using a BD FACSCalibur platform (BD Biosciences, San Jose, CA). PI was collected on a linear scale at a low flow rate. 10000 single cells were run for each sample.

## Limited proteolysis of IFT80

Serial dilutions (100 µg/ml, 10 µg/ml, and 1 µg/ml) of four proteases (elastase, trypsin, GluC, chymotrypsin) were prepared in protase dilution buffer (20 mM HEPES pH7.5, 50 mM NaCl and 10 mM MgSO$_4$). Full-length CrIFT80 was diluted to a concentration of 0.8 mg/ml in protease dilution buffer. 10 µl of the diluted CrIFT80 solution were mixed with 10 µl of 2x SDS loading buffer and kept as an 'input' sample. For each of the 12 protease dilutions, 10 µl of diluted CrIFT80 solution were mixed with 3 µl of diluted protease, and the mixture incubated on ice for 35 min. Finally, 13 µl of 2xSDS buffer was added to each tube, and the proteolysis products and input sample separated by SDS-PAGE (15% acrylamide). The resulting gel was stained with Coomassie. The GluC protease was then chosen for a small scale time-course experiment to determine the minimal time required for the production of stable fragments from a concentrated CrIFT80 stock solution. 15 µl of CrIFT80 protein (20 mg/ml in IFT80 SEC buffer) was mixed with 1 µl of a 1 mg/ml stock solution of the protease, and incubated on ice. At various time points (every 10 min for the first hour, and every 20 min for the second hour), 1 µl was removed and added to a tube containing 20 µl of 1x SDS loading buffer. At the end of the time course, the products were resolved by SDS-PAGE, which revealed 60 min as an ideal time point. For large-scale production of the stable GluC-proteolysis product for mass-spectrometric analysis as well for crystallization trials, 300 µl of CrIFT80 stock solution (20 mg/ml in IFT80 SEC buffer) were mixed with 20 µl of GluC (1 mg/ml), incubated for 60 min on ice, and then loaded immediately onto a HiLoad200 SEC column pre-equilibrated in IFT80 SEC buffer.

## Crystallization of IFT80 constructs

The proteolysed CrIFT80 construct (residues 1–654) was crystallized using the hanging drop vapour diffusion method by mixing a 500 nl of a 10 mg/ml solution of the protein in IFT80 SEC buffer with an equal volume of a precipitant containing 0.1 M MES buffer pH6.5% and 25% PEG400. Crystals typically appeared after 4–5 days at 4°C and were cryo-protected in mother liquor supplemented with PEG400 to a final concentration of 35%, and subsequently flash-cooled in liquid nitrogen. In order to obtain phase information, crystals were soaked for 5 min in mother liquor containing 1 mM ethyl-mercury phosphate (EMP) before cryoprotection and flash-cooling.

The full-length CrIFT80 protein was crystallized using the sitting drop vapour diffusion method by mixing 100 nl of a 8 mg/ml solution of the protein with an equal volume of precipitant containing 50 mM MES pH 6.0 and 100 mM Na-oxalate. Crystals typically appeared after 7–10 days incubation at 18°C. Crystals were cryoprotected in mother liquor supplemented with 30% glycerol, and subsequently flash-cooled for data collection.

## X-ray diffraction data collection and crystal structure determination

X-ray diffraction data were collected at the Swiss Light Source (SLS; Villigen, Switzerland) at the PXII beamline, and indexed with the XDS package (*Kabsch, 2010*) before scaling with Aimless as part of the CCP4 package (*Winn et al., 2011*). The structure of the proteolysed (1-654) fragment of IFT80 was determined from EMP-derivatized crystals. Single anomalous dispersion (SAD) data were recorded at the Hg peak wavelength, and AUTOSOL as part of the PHENIX package (*Adams et al., 2010*) was used to locate Hg atoms, as well as to calculate experimental phases and electron density. A partial model resulting from this work was used for molecular replacement in the program Phaser (*Storoni et al., 2004*) with the native high-resolution dataset obtained from crystals of full-length CrIFT80. The structure was finished by iterative cycles of model building in Coot (*Emsley et al., 2010*) and refinement in PHENIX. Data and refinement statistics are listed in *Table 1*.

## Interaction analysis using GST-affinity pull-downs

Tagged proteins or protein complexes were immobilized on GSH-affinity resin by incubating 200 µl of 10 µM GST-tagged complex with 15 µl bed volume of resin (volumes correspond to one pull-down reaction). As a control for non-specific interaction of the untagged protein with the affinity matrix, the same volume of resin was incubated only with buffer. Beads were harvested by centrifugation (500 g, 3 min, 4°C) after 2 hr of binding, the supernatant was discarded, and the resin washed once with binding buffer (10 mM HEPES pH 7.5, 150–250 mM K-acetate, 5% glycerol). The untagged protein was diluted to 10 µM in binding buffer and centrifuged for 5 min (12000 g, 4°C) to remove any precipitate. After removal of an 'input' sample, 200 µl aliquots of the remaining solution were

added to each tube with resin (either empty or pre-loaded with tagged protein). These mixtures were then incubated for 4–5 hr at 4°C, and the beads collected by centrifugation and washed 2–3 times with binding buffer. Finally, bound material was eluted with binding buffer supplemented with 30 mM reduced glutathione. Inputs and eluates were compared by SDS-PAGE.

## Interaction analysis using isothermal titration calorimetry (ITC)

ITC was carried out using a Microcal PEAQ-ITC calorimeter (Malvern). All proteins were in a buffer containing 10 mM HEPES pH 7.5, 100 mM NaCl, and 5% glycerol. A volume of 200 µl of IFT80(BP1-BP2) protein (concentration 20 µM) was titrated with WT or mutant IFT38CH (concentration 200 µM) at 25°C. For each ITC curve, a background curve consisting of the titration of IFT38CH into buffer was subtracted to account for heat of dilution. The ITC data were analysed using the program Micro-Cal PEAQ-ITC provided by Malvern.

## Small angle X-ray scattering

SEC-SAXS experiments were performed at the BM29 beamline (ESRF, Grenoble, France) equipped a Pilatus 1M detector using a similar protocol the one used previously (*Brennich et al., 2017*). In brief, SAXS data were collected on proteins and protein complexes eluting directly from a Superdex 200 10/300 GL SEC column. For the concentration series measurements, 30 µL of protein sample ranging from 0.4 to 8 mg/ml was used and 10 scans were measured for each sample. Buffer was measured before and after each sample measurement. Radiation damage, data merging and buffer subtraction were performed on site and later verified manually using the program PRIMUS (*Konarev et al., 2003*). All SAXS parameters such as maximum particle size (Dmax) were extracted using GNOME from the ATSAS package software (*Petoukhov et al., 2012*). Theoretical SAXS curves were calculated from crystal structures using CRYSOL (*Svergun et al., 1995*) and fitted to the experimental data. *Ab initio* models of protein envelopes were calculated using the bead-modeling program DAMMIN to generate ten independent reconstructions. SUPCOMB and DAMAVER were employed to generate the average representative models shown in *Figure 5C–D*.

## Dynamic light scattering (DLS) assays

DLS measurements were performed using a DynaPro-MS/X (Wyatt Technologies, Santa Barbara, CA) system. 10 µL of protein sample were loaded into a 1.5 mm path length, disposable cuvette. Data were acquired using a Wyatt DynaPro Nano Star DLS instrument and analyzed with the Dynamics software. The Stokes-Einstein equation was used to calculate the hydrodynamic radii of the particles from four independent experiments each composed of 15 separate acquisitions. The data were plotted using the program Prism 7.

## Plasmids

Plasmids pPB-RN and pPB-T-PAF (*Li et al., 2013*) were kindly provided by Prof. J. Rini (University of Toronto, Canada) and Dr. A. Nagy (Lunenfeld-Tanenbaum Research Institute, Toronto, Canada), pBase (pCyL43) by the Wellcome Trust Sanger Institute (Hinxton, UK).

## Cell culture

IMCD cells were maintained in DMEM-F12 (Lonza, Switzerland), with Glutamine, 10% fetal calf serum (FCS, Gibco), 100 IU/ml penicillin and 100 µg/ml streptomycin (Thermo Fisher Scientific). IMCD3 cells were transfected with pRN, pBase and pPB-T encoding the respective GFP-labelled human IFT 80 constructs in 1:1:11 ratio using Lipofectamine 2000 (Thermo Fisher Scientific) and maintained in selection medium containing 0.5 mg/ml G418 (Merck, Germany) and 0.25 µg/ml puromycin (Merck) for 3 weeks. Cell lines were authenticated by STR profiling and were mycoplasma free.

## Immunostaining and imaging of cells

For ciliogenesis, IMCD3 cells were grown to confluence on glass cover slips, starved for 48 hr in DMEM-F12 without FCS and 1 µg/ml doxycycline, fixed with 4% formaldehyde, washed with PBS and Tris buffer (100 mM Tris, pH = 7.4, 50 mM NaCl), permeabilized with 0.5% Triton X-100 in PBS for 10 min, blocked in PBS, pH = 7.4, 1% BSA, 2% FCS for 30 min, incubated with Anti-Acetylated Tubulin antibody (1:1000, Sigma, T7451) antibody for 2 hr, washed twice with PBS, incubated with

Alexa-594-labelled Anti-mouse antibody (1:500, Thermo Fisher Scientific), stained with 1 µg/ml DAPI (Thermo Fisher Scientific) for 15 min and washed three times with PBS. Glass slides were mounted in Mowiol (Merck) and imaged on a Zeiss LSM 710 confocal microscope using the ZEN software. For quantification, stacks of 3 images were recorded in four random areas for each biological replicate, using a 40x/1.2 water immersion objective in the DAPI, GFP and Alexa 594 channel. Number of cilia and nuclei were counted from the Maximum Intensity Projection images of each channel manually, randomised, and blinded. Midbodies were excluded from the quantification. All n values represent biological replicates. For representative images, stacks of 8 images were recorded using a 63x/1.4 oil immersion objective in the DAPI, GFP and Alexa 594 channel and Maximum Intensity Projection images were further processed using ImageJ (*Schneider et al., 2012*).

## In-Gel GFP fluorescence measurements

IMCD3 parental and IFT80-ko cells stably expressing the indicated constructs were grown to confluency in 6-well plates in DMEM-F12 with 10% FCS (with the addition of 0.5 mg/ml G418 and 0.25 µg/ml puromycin for stable cells) for 48 hr and starved and induced for 48 hr in DMEM-F12 without FCS containing 1 µg/ml doxycycline (and 0.5 mg/ml G418, 0.25 µg/ml puromycin for stable cells). Cells were lysed in 180 µl RIPA buffer (20 mM Tris/HCl, pH 7.4, 150 mM NaCl, 3 mM EDTA, 2% TX-100) with 1x complete protease inhibitor mix (Merck). 4–12 µg lysate were loaded onto 12.5% SDS gels without boiling of the sample and GFP fluorescence was detected in-gel on a Typhoon Trio (Amersham Biosciences/GE Healthcare) using 488 nm excitation and a 534/40 emission filter. The GFP fluorescence intensity was quantified using ImageJ and normalised by total protein amount in each lane quantified from coomassie staining.

## Super-resolution microscopy and automated quantification of cilia staining

A Zeiss LSM 880 Multiphoton upright confocal microscope with Airyscan was used for super-resolution microscopy. Images were acquired with a 63x/1.4 Oil immersion PlanApochromat objective, and optimal airyscan setting in super-resolution mode. To quantify cilia length and proportions of GFP +cilia automatically over many cells and fields of view, images were acquired at 40X magnification using standard fluorescence imaging. We wrote the macros presented in supplementary material to quantify cilia using Arl13b / γ-tubulin/GFP immunostaining. The macros that we wrote to automatically measure cilia lengths (Arl13b and acetylated tubulin staining) or numbers of IFT80-GFP spots within cilia are freely available at the following links: https://www.ucl.ac.uk/ich/core-scientific-facilities-centres/confocal-microscopy/documents/Cilia_lengths_2D.ijm; https://www.ucl.ac.uk/ich/core-scientific-facilities-centres/confocal-microscopy/documents/Cilia_Spots_2D.ijm

## Acknowledgements

We would like to thank the crystallization facility at the Max-Planck Insitute of Biochemistry (MPIB) for invaluable help with crystallization screening, and the staff at the Swiss Light Source (SLS) at the Paul-Scherrer-Institute (PSI) in Villigen, Switzerland, for help with crystal data collection, and the staff at beam line BM29 at the ESRF, Grenoble, France, for help with SAXS acquisition as well as Ralf Stehle for assistance with initial SAXS measurements. Furthermore we are grateful to Elfriede Eppinger in the Department of Structural Cell Biology at the MPIB for help with insect cell culture, and to Lissy Weyher and Dr. Stephan Übel in the MPIB core facility for mass-spectrometry and help with biophysical protein analysis, respectively. We also thank Lene Heegaard Madsen and Marcin Nadzieja at Aarhus University for access to confocal microscopy and Anni Christensen for technical assistance with western-blotting. This work was funded by a project grant from the Novo Nordisk Foundation (grant number NNF15OC00114164) to EL, and by a New Investigator Research Grant (MR/L009978/1) and BBSRC/CASE studentship (STU100044631) to DJ. This research was also supported by the National Institute for Health Research Biomedical Research Centre at Great Ormond Street Hospital for Children NHS Foundation Trust and University College London.. Structural coordinates have been deposited at PDB, accession code 5N4A.

## Additional information

### Funding

| Funder | Grant reference number | Author |
|---|---|---|
| Novo Nordisk | NNF15OC00114164 | Esben Lorentzen |
| Medical Research Council | MR/L009978/1 | Dagan Jenkins |
| Biotechnology and Biological Sciences Research Council | STU100044631 | Dagan Jenkins |

The funders had no role in study design, data collection and interpretation, or the decision to submit the work for publication.

### Author contributions

Michael Taschner, Conceptualization, Data curation, Formal analysis, Validation, Investigation, Visualization, Methodology, Writing—original draft, Writing—review and editing; Anna Lorentzen, Conceptualization, Data curation, Formal analysis, Investigation, Visualization, Methodology, Writing—original draft, Writing—review and editing; André Mourão, Data curation, Formal analysis, Investigation, Visualization, Methodology; Toby Collins, Conceptualization, Investigation, Visualization, Methodology; Grace M Freke, Jerome Basquin, Investigation, Methodology; Dale Moulding, Software, Formal analysis, Methodology; Dagan Jenkins, Conceptualization, Data curation, Formal analysis, Supervision, Funding acquisition, Validation, Investigation, Visualization, Methodology, Writing—original draft, Project administration, Writing—review and editing; Esben Lorentzen, Conceptualization, Data curation, Formal analysis, Supervision, Funding acquisition, Validation, Investigation, Methodology, Writing—original draft, Project administration, Writing—review and editing

### Author ORCIDs

Esben Lorentzen (iD) http://orcid.org/0000-0001-6493-7220

### Decision letter and Author response

Decision letter https://doi.org/10.7554/eLife.33067.029
Author response https://doi.org/10.7554/eLife.33067.030

## Additional files

### Supplementary files

• Transparent reporting form
DOI: https://doi.org/10.7554/eLife.33067.024

### Major datasets

The following dataset was generated:

| Author(s) | Year | Dataset title | Dataset URL | Database, license, and accessibility information |
|---|---|---|---|---|
| Michael Taschner, André Mourão | 2017 | Crystal structure of Chlamydomonas IFT80 | https://www.rcsb.org/structure/5N4A | Publicly available at the RCSB Protein Data Bank (accession no. 5N4A) |

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
