## [Decision Letter]

Thank you for submitting your article "Crystal structure of intraflagellar transport protein 80 reveals a homo-dimer required for ciliogenesis" for consideration by *eLife*. Your article has been favorably evaluated by Andrew Musacchio (Senior Editor) and three reviewers, one of whom is a member of our Board of Reviewing Editors. The reviewers have opted to remain anonymous.

The reviewers have discussed the reviews with one another and the Reviewing Editor has drafted this decision to help you prepare a revised submission.

The authors of this study combine structural and biophysical techniques to characterise the IFT80 subunit of the Intra-flagellar Transport (IFT) B complex. Crystal structures of full-length IFT80 reveal high structural similarity between the first β-propeller and coatamer subunits. The second β-propeller assumes an unusual open conformation, which, together with an extended C-terminal α helix, forms the interface for IFT80 homo-dimerization. Using a number of biophysical methods and deletion analyses, they present data to suggest that IFT80 predominantly exists as a dimer in solution.

In order to understand how IFT80 is incorporated into the IFT-B2 complex, the authors investigate its binding to IFT38. By pull-down experiments of full length IFT80 and several truncated versions of the protein, they show that the IFT80 β-propeller 1 tethers IFT80 to the IFT complex through a high affinity interaction with IFT38.

The authors also use CRISPR/Cas to show that IFT80 is absolutely required for initiation of ciliary axoneme assembly and IFT80 homo-dimerization is required for cilium formation in mammalian cells. They conclude by suggesting that IFT80 is responsible of the dimerization of the complete complex B in the IFT train structure.

The reviewers were generally positive about the paper and felt that it is appropriate for *eLife* as it represents an important structural advance towards the structure of the IFT particle. However there are some points that need addressing before it can be accepted.

Major comments:

1) It is important to ascertain by immunoblotting that mutant alleles generated by CRISPR represent complete protein nulls for IFT80. Also, it is important to check the protein expression levels of the missense mutants. In the rescues of IFT80ko cells with the various GFP-IFT80 constructs, the authors need to compare the levels of each GFP fusion with respect to one another by quantitative western blotting with an anti-GFP antibody and with respect to the endogenous IFT80 using and anti-IFT80 antibody. This will confirm the 'hypomorphic' nature of the pathogenic mutations and strengthen their conclusion that these mutations likely destabilise IFT80. This type of experiment becomes particularly important when the authors want to claim that 'The reduction of cilium formation in the IFT80 1-657 rescue experiment is likely a result of the reduced ability of this construct to bind IFT54/20'.

2) The IFT80 structure is minimally leveraged when the authors are attempting to test the importance of dimerization on ciliogenesis. While reasonable, the argument that 'Point mutations of IFT80 dimerization interface residues did not break up the homo-dimer, presumably because of the highly extended interface containing many backbone-backbone interactions (data not shown)' suggests that multiple mutations need to be introduced at once. The authors should generate and test a multiply substituted variant of IFT80 defective in dimerization.

3) 'In SEC experiments, IFT80 eluted as a dimer at low salt conditions (150 mM NaCl), but as a monomer at higher salt concentrations (500 mM NaCl, Figure 5A).' A proper analysis of the elution peaks compared to size markers, void volume and excluded volume needs to be presented.

4) There needs to be stronger evidence that IFT80 dimers play a role in train assembly and not just IFT-B complex assembly. If IFT80 was responsible for the dimerization of complex B for train formation, the authors should be able to obtain such a dimer from in vitro reconstituted complex B particles (Taschner et al., 2016). The authors should have all the reagents and tools to address this point. For instance, they could perform SAXS or negative-stain EM on reconstituted complex B (15 proteins) and complex B2 (6 proteins) and check for the presence of dimers. Such an experiment would provide convincing evidence for their hypothesis. The authors should otherwise reformulate their conclusions and provide alternative interpretation for the role of IFT80 dimerization in the assembly of IFT trains.

---

## [Author Response]

Major comments:1) It is important to ascertain by immunoblotting that mutant alleles generated by CRISPR represent complete protein nulls for IFT80. Also, it is important to check the protein expression levels of the missense mutants. In the rescues of IFT80ko cells with the various GFP-IFT80 constructs, the authors need to compare the levels of each GFP fusion with respect to one another by quantitative western blotting with an anti-GFP antibody and with respect to the endogenous IFT80 using and anti-IFT80 antibody. This will confirm the 'hypomorphic' nature of the pathogenic mutations and strengthen their conclusion that these mutations likely destabilise IFT80. This type of experiment becomes particularly important when the authors want to claim that 'The reduction of cilium formation in the IFT80 1-657 rescue experiment is likely a result of the reduced ability of this construct to bind IFT54/20'.

To address this point, we have done Western blotting using two commercially available antibodies against IFT80, however, neither of these gave a clean result using a variety of conditions (Author response image 1). We also found that these antibodies did not work well on immunofluorescence.

**Author response image 1. respfig1:** IFT80 Western blots. Western blots performed on lysates of parental and *Ift80null* mIMCD-3 cells expressing human truncation mutants using various conditions and antibody concentrations could not detect mouse or human IFT80. Two representative Western blots are shown using anti-IFT80 antibodies #25230-1-AP from ProteinTech (**A**) or #PAB15842 from Abnova (**B**). The expected molecular weight of endogenous human IFT80, which should be expressed in the parental cell line (left lane) is indicated by a red arrowhead.

However, several lines of evidence support that the frameshift mutations are indeed null alleles. Firstly, we have demonstrated that the p.C288fsX and p.G289fsX mutations result in considerable levels of transcript instability (Figure 7—figure supplement 2). By contrast the frameshift mutations at residues p.T23, p.T24 and p.V20 did not lead to transcript instability – this may reflect the use of an internal translation initiation codon. As shown in Figures 3B and 9A, all frameshift mutations will lack the IFT80 dimerisation domain as well as the α helical extension required for IFT54/20 binding, while any protein generated through use of an internal translation initiation complex would lack BP1 essential for IFT38 binding. Finally, we showed in Figure 9B that a construct with a truncation at residue 587 was non-functional, failing to rescue ciliogeness. As such, it is likely that the frameshift mutations we created using CRISPR are also functional null alleles.

As suggested by the reviewers, we have also monitored the expression levels of mIFT80-GFP wild-type and missense mutant constructs and also the GFP-IFT80 truncation constructs using anti-GFP Western blotting and in-gel fluorescence (new Figure 3—figure supplement 1D-E and Figure 8—figure supplement 3). This analysis shows that all missense mutant forms of mIFT80-GFP produce specific bands at 106.9 kDa. This band was not detected in mock-transfected cells, or in cells transfected with empty vector, in which GFP was detected at 26.9 kDa. We noted that both of these bands were considerably diminished in intensity for all missense mutant transfections, as compared to wild-type. This showed that significant levels of all missense mutant forms of mIFT80-GFP were expressed, but that the levels were much lower than for wild-type. Therefore, IFT80 missense mutations are likely to cause protein instability. When considered in combination with the rescue data, this supports the contention that these mutations are hypomorphic. Using in-gel fluorescence, we also found that all truncation forms of GFP-IFT80 studied in Figure 9 were expressed at similar levels to wild-type protein showing that the failure of the dimerization domain deletion mutant to rescue ciliogenesis represents loss of protein function rather than reduced protein production (new Figure 3—figure supplement 1D-E).

We have added the Western blot data for transfected mIFT80-GFP constructs in a new Figure 8—figure supplement 3, and described the result as follows: ‘Western blotting showed that all transfected missense mutant forms of mIFT80-GFP were expressed at significantly lower levels that wild-type, suggesting that these mutations cause protein instability’.

2) The IFT80 structure is minimally leveraged when the authors are attempting to test the importance of dimerization on ciliogenesis. While reasonable, the argument that 'Point mutations of IFT80 dimerization interface residues did not break up the homo-dimer, presumably because of the highly extended interface containing many backbone-backbone interactions (data not shown)' suggests that multiple mutations need to be introduced at once. The authors should generate and test a multiply substituted variant of IFT80 defective in dimerization.

We agree with the reviewers’ comments that point-mutations resulting in a break-up of the dimer are indeed preferable and this was indeed our initial approach. However, as we state in the manuscript, single point mutations did not result in dimer disruption as analysed by SEC/SAXS and DLS. This was both the case for mutations to alanines as well as to larger more bulky residues. Constructs where several residues were simultaneously mutated were also cloned but these did not express in small-scale tests using insect cells. We think the reasons for these observations are not only that the interface is quite extended but also that it contains many sequence-unspecific backbone interactions as illustrated by β-sheet formed between blades of the second β-propeller in the dimeric structure. We thus had to rely on the more crude approach of truncating the IFT80 protein to address the functionality of the dimerization domain.

3) 'In SEC experiments, IFT80 eluted as a dimer at low salt conditions (150 mM NaCl), but as a monomer at higher salt concentrations (500 mM NaCl, Figure 5A).' A proper analysis of the elution peaks compared to size markers, void volume and excluded volume needs to be presented.

Figure 5A has been improved and now includes Mw for each peak as calculated from the elution of spherical standard Mw markers for this column. Given that SEC is a shape-dependent method, Mw values will deviate from the theoretical values (now mentioned specifically in the figure legend). Also, a colour code consistent with Figure 5B has been introduced. All elution peaks are far from the void volume (45 mL, now mentioned in the figure legend) but we prefer not to show the entire elution profile (0-120mL) because of space restrictions and the fact that the four IFT80 elution peaks would cluster very close together.

4) There needs to be stronger evidence that IFT80 dimers play a role in train assembly and not just IFT-B complex assembly. If IFT80 was responsible for the dimerization of complex B for train formation, the authors should be able to obtain such a dimer from in vitro reconstituted complex B particles (Taschner et al., 2016). The authors should have all the reagents and tools to address this point. For instance, they could perform SAXS or negative-stain EM on reconstituted complex B (15 proteins) and complex B2 (6 proteins) and check for the presence of dimers. Such an experiment would provide convincing evidence for their hypothesis. The authors should otherwise reformulate their conclusions and provide alternative interpretation for the role of IFT80 dimerization in the assembly of IFT trains.

We agree with this point and have carried out additional experiments with an IFT80 containing 14-subunit IFT-B complex. We left out IFT172 from the reconstitution as this protein tends to dissociate from the IFT-B complex at lower concentrations, which would make the DLS/SAXS data hard to interpret. Both DLS and SAXS data demonstrate a concentration-dependent increase in size of the complex consistent with a monomer-dimer equilibrium. The data have been included in a new Figure 6 and are discussed in the subsections “IFT80 dimerization requires the first residues of the C-terminal α-helical extension” and “Concentration dependent multimerization of a reconstituted 14-subunit IFT-B complex”.